# Age-Related Macular Degeneration (AMD): Pathophysiology, Drug Targeting Approaches, and Recent Developments in Nanotherapeutics

**DOI:** 10.3390/medicina60101647

**Published:** 2024-10-08

**Authors:** Mahendra Singh, Riyakshi Negi, Ramachandran Vinayagam, Sang Gu Kang, Prashant Shukla

**Affiliations:** 1Department of Biotechnology, Institute of Biotechnology, School of Life and Applied Sciences, Yeungnam University, Gyeongsan 38541, Republic of Korea; m.singh2685@gmail.com; 2Department of Pharmaceutical Sciences, School of Heath Sciences and Technology, UPES, Dehradun 246008, India; riyakshi.107516@stu.upes.ac.in (R.N.); alkam.bbau@gmail.com (A.)

**Keywords:** age-related macular degeneration, gene therapy, oxidative stress, inflammation, chronic disease, nanomedicine, anti-VEGF agents

## Abstract

The most prevalent reason for vision impairment in aging inhabitants is age-related macular degeneration (AMD), a posterior ocular disease with a poor understanding of the anatomic, genetic, and pathophysiological progression of the disease. Recently, new insights exploring the role of atrophic changes in the retinal pigment epithelium, extracellular drusen deposits, lysosomal lipofuscin, and various genes have been investigated in the progression of AMD. Hence, this review explores the incidence and risk factors for AMD, such as oxidative stress, inflammation, the complement system, and the involvement of bioactive lipids and their role in angiogenesis. In addition to intravitreal anti-vascular endothelial growth factor (VEGF) therapy and other therapeutic interventions such as oral kinase inhibitors, photodynamic, gene, and antioxidant therapy, as well as their benefits and drawbacks as AMD treatment options, strategic drug delivery methods, including drug delivery routes with a focus on intravitreal pharmacokinetics, are investigated. Further, the recent advancements in nanoformulations such as polymeric and lipid nanocarriers, liposomes, etc., intended for ocular drug delivery with pros and cons are too summarized. Therefore, the purpose of this review is to give new researchers an understanding of AMD pathophysiology, with an emphasis on angiogenesis, inflammation, the function of bioactive lipids, and therapy options. Additionally, drug delivery options that focus on the development of drug delivery system(s) via several routes of delivery can aid in the advancement of therapeutic choices.

## 1. Introduction

The aging process has attracted human curiosity since the emergence of human history. But its intricacy has rendered accurate understanding and thus, an appropriate explanation is difficult. Currently, aging is described by some researchers as “the process of accumulation of consequences of life, such as molecular and cellular damage, that leads to functional decline, chronic diseases, and ultimately mortality” [1]. Accordingly, as aging is an inevitable result of life, it can be claimed that it is a physiological process. It explains why this process is so common in living things, as it is most likely related to the molecular damage that occurs during life [2]. The biological hallmarks of aging are genomic instability, telomere attrition, epigenetic changes, loss of proteostasis, impaired macroautophagy, deregulated nutrition sensing, mitochondrial dysfunction, stem cell exhaustion, cellular senescence, altered intercellular communication, chronic inflammation, and dysbiosis [3]. It is significant to note that these systems are modifiable through therapeutic interventions and are interdependent. Two primary categories of theories—programmed or non-stochastic and non-programmed or stochastic aging theories have been developed to address this subject [2]. According to the theories of programmed aging, aging has been selected for by evolution for a variety of reasons. For example, some hypotheses indicate that aging originated as a control of the population’s size and for the benefit of the species. The non-programmed aging theories, on the other hand, argue that aging is a natural consequence of life and the outcome of damage accumulating over time.

Furthermore, aging has been documented to exert a significant influence on the initiation and progression of numerous diseases, including cancer, cardiovascular conditions, and eye diseases. Visual impairment and blindness, especially among the elderly, profoundly affect the quality of life, leading to diminished physical mobility, heightened risks of depression and anxiety, and increasing global healthcare costs [4,5,6]. Age-related macular degeneration (AMD) in elderly patients has arisen as one of the obvious triggers of blindness remarkably in developed, and prevalent in developing countries [7]. It accounts for 8% of blindness globally and is a leading cause of irreparable vision damage among people aged 60 years and above [4].

AMD disease usually arises due to complex changes developing because of aging, genetic and environmental risk factors, and changes in protein homeostasis. These variations along with age-related changes in anatomy and physiology have led to impaired vision.

It is the first major review that will cover various topics including etiology, Types and Symptoms, prevalence and risk factors; pathogenesis focusing on molecular aspects, particularly inflammatory, oxidative, and angiogenesis aspects; therapeutic intervention in AMD; strategic drug delivery approaches. Additionally, a summary of therapies or drug delivery systems in experimental and clinical settings, as well as the benefits and drawbacks of drug delivery approaches for AMD treatment, will be covered. Hence, this review will be valuable to new researchers who are working or going to work in the field of AMD.

## 2. Etiology, Types and Symptoms of Age-Related Macular Degeneration

Age-related macular degeneration or AMD can be described as a chronic, complex disease that occurs due to multiple genetic variants like the Complement factor H-associated gene, environmental factors like exposure to cigarette smoke and UV radiation, and lifestyle choices such as smoking [8,9,10,11]. The AMD can be classified based on clinical characteristics into dry (non-vascular or early stage) AMD and wet AMD (neovascular or late stage) characterized by neovascularization in the macula [12,13]. In most cases, progressive loss of vision starts with dry AMD, and in about 10% of the patients it is converted into more aggressive wet macular degeneration with neovascularization as a key feature in the macular region. Although 90% of cases are associated with dry AMD, there is no clinical therapy and clinically is managed by suggesting the patient to evade smoking, follow a healthy lifestyle, and have antioxidant supplements. In the case of wet AMD, the current clinical therapies are based on targeting vascular endothelial growth factor (VEGF) by recently approved anti-VEGF antibodies [14]. The delivery of these antibodies to the target site is a major task due to poor permeability after topical administration leading to multiple intravitreal injections for therapy which leads to a reduction in patient compliance and increases the risk for intraocular infections [15,16].

### Prevalence and Risk Factors of AMD

Age-related macular degeneration (AMD) is a disorder correlated with the age of the population. Hence, the prevalence is extremely dependent upon the age-related demographics in the world. It is projected that the number of patients suffering from AMD may rise to 288 million by 2040 [17,18]. Currently, the majority of AMD patients are in regions having a higher number of Aging population but in the future, the demographics can shift to the countries in the Asian region, and is expected to reach 113 million cases by 2040 [19]. Europe is projected to be second to Asia in the number of expected cases (69 million in 2040), followed by Africa (39 million), Latin America and the Caribbean (39 million), North America (25 million), and Oceania (2 million) [19].

AMD burdens of African and Eastern Mediterranean regions were significantly higher than American, South-East Asian, European, and Western Pacific regions [20]. Frank et al. reviewed the literature and have presumed that AMD is much more common in white persons than in persons of black African inheritance [21]. A cross-sectional study using data from the Korea National Health and Nutrition Examination Survey (KNHANES) indicated that the prevalence of age-related macular degeneration (AMD) in South Korea was 13.94% among the cohort of participants aged 40 and older [22]. In the case of the Indian subpopulation, the overall prevalence was found to be in the range of 1.4% (in the Western region) and 3.1% (in the Southern region). The prevalence of AMD (11.64%) was slightly lower than in the South Korean population with a higher prevalence in the white population as compared to the black population.

Age along with other factors, for instance, race blood pressure, and daily routine have additionally constituted the risk factors for the progress of AMD [23]. Family history of AMD, cataract surgery, and smoking were furthermore found important clinical threat factors for AMD [24].

Higher body mass index, cardiovascular disorders along with hypertension, and increased plasma fibrinogen were found to have a moderate and consistent association with AMD as demonstrated by meta-analysis outcomes [24]. Genetic issues have a strong impression on the pathogenesis of AMD [19]. The gene related to lipid transport or metabolism (APOE), complement activation (CFH and FTRA1), angiogenesis, and apoptosis is involved in wet AMD, whereas only one gene i.e., TLR3 is related to dry AMD and is tangled in apoptosis of retinal pigment epithelium (RPE) by innate immunity (Figure 1) [23].

## 3. Pathologies Associated with AMD

The AMD is a catastrophic eye illness that damages the macula, the key region of the retina accountable for clear and sharp vision [25,26]. In the initial stages of AMD, visual impairment may be minimal, and patients may experience decreased reading ability, visual distortion (such as straight lines appearing wavy), or the presence of a dark or gray spot in the central vision [27]. Patients may experience a moderate to significant increase in the size of vitreous wart membranes or partial or incomplete macula pigmentation as the disease advances to the intermediate stage. This can cause a greater degree of visual impairment, including difficulty with fine details, color perception, and contrast sensitivity [26]. In the advanced stage of AMD, the central vision is significantly affected, and patients may experience a large blind spot or distortion in the center of their vision, making it tough to perform routine activities such as reading, driving, or identifying faces. In some cases, patients may have only peripheral vision remaining. Age-related changes in the eye are often initiated by the accumulation of uncleared cellular debris (drusen a clinical hallmark of AMD) originating from the retinal pigmented epithelium (RPE) in the region of interface between RPE, Bruch’s membrane, and the choroid [28,29]. An extracellular matrix with many layers, known as Bruch’s membrane, functions as a barrier against pathological processes like choroidal neovascularization (CNV) and as a physical and biochemical support system for normal physiological activities. Drusen is composed of cellular proteins, lipids, and carbohydrates and finally forms white or yellow deposits in the macular region. This process along with concomitant processes such as oxidative stress, angiogenesis, inflammation, and activation of complement cascade results in reduced permeability of thickened Bruch’s membrane culminating in reduction of transport of nutrients to RPE and waste product to choroid in addition to reduced choroidal vasculature [30,31,32,33].

### 3.1. Oxidative Stress

The difference between the tissues’ generation of reactive oxygen species (ROS) and their capacity to remove these ROS from the body and repair the damage they have caused is known as oxidative stress [31,34]. Although ROS are typical byproducts of biological metabolism, they can harm cells, including those in the retina, when they build up too much. Furthermore, oxidative stress damages the retina due to exposure to light, high oxygen consumption, and low antioxidant capacity [30]. It has been proved that oxidative stress in AMD damages the RPE, a layer of cells that supports the unique function of photoreceptor cells in the retina. The gathering of damaged RPE cells, and the formation of drusen, small yellow deposits that form beneath the retina, are hallmarks of early AMD [20].

Aging contributes to a great extent due to a decrease in plasma glutathione levels and an increase in concertation of the oxidized form of glutathione [26,35]. Furthermore, aging leads to other changes such as reduced vitamin A, catalase activity, and reduction in macular pigment optical density. In addition, the presence of lipofuscin generates ROS leading to RPE cell death via compromising lysosomal integrity, and lipid peroxidation causes RPE cell death [36,37].

Moreover, inflammation brought on by oxidative stress is believed to have a role in the advancement of AMD. Inflammatory cells and molecules have been found in the retina of individuals with AMD, and these can further exacerbate oxidative stress. Several hereditary variables have been shown to influence a person’s vulnerability to oxidative stress and AMD development (Figure 1). These genes are involved in the control of the pathways leading to oxidative stress and inflammation. To sum up, oxidative stress is a key reason for the onset and advancement of AMD. A deeper comprehension of the mechanisms underlying oxidative stress together with the genetic and environmental factors that contribute to it may lead to novel treatments for this excruciating sickness.

### 3.2. Inflammation

Lipids and lipoproteins alone cannot contribute to inflammation triggering AMD pathobiology which can lead to drusen generation in the macula [38]. The oxidized lipids including, 4-hydroxynonenal carboxyethyl pyrrole, oxidized phospholipids, isolevuglandins, malondialdehyde, and 7-ketocholesterol [11] play a significant role in the generation of the highly stressed environment in the macula. Macula can be a trigger for oxidative stress where antioxidant defenses are reduced e.g., in case of aging [39]. The oxidized lipids can react with the other biomolecules such as proteins can lead to a generation of neoepitopes that can be recognized by and removed by immune cells, but in a case where there is an excessive generation of these neoepitopes and impaired removal by immune cells due to aging or genetic factors such as CHF 402H gene can proceed to pro-inflammatory immune response and macrophage/microglia recruitment leading to RPE degeneration [11].

Retinal microglia (i.e., immune cells) play a key role in initiating retinal inflammation. Several retinal disorders, such as glaucoma, AMD, and diabetic and hereditary retinopathy, have been linked to aberrant microglial activation. Increased pro-inflammatory cytokines and NLRP3 inflammasome activation may result from the presence of enlarged amoeboid microglia in AMD retinal sections next to RPE cells covering drusen [40]. This could lead to reactive microglial cells causing photoreceptor cell death and retinal degeneration, and the infiltration of microglia to the retina may have an impact on the integrity of the neural retina. Furthermore, the precise role that invading macrophages play in retinal illnesses may vary depending on how well they remove debris and eliminate immune cells from the body. Overall, these findings highlight how critical it is to understand the roles played by macrophages and microglia in retinal illnesses in order to design effective treatments.

### 3.3. Angiogenesis

Angiogenesis plays a crucial part in AMD’s progression from the disease’s early (dry form) to its later, more severe (wet form). Wet AMD is described by malformed blood vessels that protrude underneath the retina, damaging the RPE and photoreceptor cells and quickly impairing vision. The development of these malformed blood vessels is fuelled by VEGF, which is produced in response to ischemia (lack of oxygen) and other factors such as inflammation [23,41,42,43]. Besides, VEGF induces neovascularization and encourages endothelial cell migration and proliferation, which forms the inner lining of blood vessels. Likewise, wet AMD causes abnormally high levels of VEGF in the eye, which promotes the formation of extra blood vessels (CNV). These blood vessels are leaky and fragile and can cause bleeding and scarring in the macula. Furthermore, it is thought that the presence of large drusen deposits initiates the conversion process by causing complement activation and VEGF and Platelet-derived growth factor (PDGF) secretion [14]. This can result in increased vascular permeability, macular edema, exudation, and CNV. As the extremely delicate capillaries deteriorate, subretinal hemorrhage, and retinal cytotoxicity may result, and as CNV ages, disc-shaped scar formation may occur, signifying an incurable end stage of wet AMD [44]. The goal of current wet AMD treatments is to inhibit the formation of aberrant blood vessels by targeting VEGF. Anti-VEGF medications, like bevacizumab, aflibercept, and ranibizumab, are injected into the eye to inhibit VEGF activity and lessen the development of aberrant blood vessels [27].

In addition to VEGF, research has looked into PDGF’s role in AMD pathogenesis. Research has shown that PDGF stimulates pericyte endothelial interactions, which stabilizes mature capillaries, whereas VEGF promotes the formation of new vessels. Furthermore, through stimulating macrophages and fibroblasts, PDGF plays a critical role in ocular fibrosis. Various other mediators responsible for angiogenesis are shown in Figure 2. The shortcomings of anti-VEGF therapy can be mitigated by using anti-PDGF therapy has already been established by numerous studies [27,45,46].

### 3.4. Complement System and AMD

The complexity of AMD pathogenesis and pathophysiology is responsible for the lack of fully effective therapeutic options. Many investigations using molecular biology and genetic studies have illustrated the role of the complement system [47,48,49,50,51]. Typically, the complement has a role in identifying infections, debris, and dead cells and mediating their removal [52,53].

The pathophysiology of AMD is significantly influenced by the functional aberrations of the complement system. The function of complement proteins that are produced locally, such as FD (Factor D), FHL-1 (a protein similar to Factor H), and complement proteins that are found throughout the body or blood that have an effect locally on tissues (such as FHRs) [54].

A highly activated complement system has been reported to be present in patients with intermediate and final stages of dry AMD. Increased age and AMD lead to the accumulation of membrane attack complex (MAC) in choriocapillaris and capillary septa. In late-stage AMD, the MAC localizes into the proximity of RPE cells [55,56]. Normally, FH (factor H) and Factor H-like proteins play a critical protagonist in the defence of ECM (extracellular matrix beneath the retina) [54,57,58].

The RPE cells having genetic polymorphism in FH 402H demonstrate a high tendency of AMD via lowered mitochondrial and phagolysosome (L) activity [52]. This leads to altered interaction of BM with integrins. In older populations, additional risk factors lead to further alterations in the Birch Membrane ECM leading to reduced glucose transport. The change in ECM additionally triggers complement turnover and further potentiated by the presence of high-risk variants in FHL1, FH, FI, and accumulated FHR proteins (i.e., FH antagonists). This leads to a loss of control over the conversion of C3 to C3a [59]. Further, the downstream products of complement turnover accumulate in the BM localizing in inter-capillary septa. the changes in permeability of ECM furthermore lead to a rise in oxidative stress in RPE cells by enhancing metabolic stress finally culminating in stimulation of inflammation [26]. Changes in ECM additionally lead to the accumulation of products of lipid peroxidation and Hydrogen peroxide (H_2_O_2_) in the altered BM, promoting the upregulation of inflammatory cytokines. This along with the presence of genetic factors (high-risk FH 402H variant) and accumulation of FHR accumulation directs to further amplification of the pathologies related to early AMD [11]. MAC accumulation in drusen further promotes inflammation. Localized inflammation, in conjunction with the release of complement mediators C3a and C5a, causes immune cell recruitment and activation, as well as mast cell degranulation, into the BM/RPE/retina interface, aggravating retina homeostasis leading to wet AMD [27].

In addition, FHR-4 which is found in the inter-capillary septa of the choriocapillaris and ECM between fenestrated capillaries of the choriocapillaris and is thought to have a role in regulating the balance between proteins that activate and inhibit the alternative pathway of the complement system, thereby modifying AMD risk [54]. In the case of AMD high systemic levels of FHR-4 may lead to increased deposition of FHR-4 in BM, drusen, and choriocapillaris which can compete with FH for C3b binding which inhibits the FI-mediated C3b activation leading to reduced clearance of debris for retina and macular region.

A study investigated that genetic factors were associated with AMD. The investigation discovered 52 genetic variants in 34 different places, including ultra-rare variants in complement factors 8A and 8B. These two components of the terminal complement (MAC) were involved in the destruction of pathogens and injured cells. The discovered C8 variations affected C8 triplex protein local interactions in vitro, suggesting their impact on MAC stability. The findings confirmed that targeting MAC rather than the initial stages of the complement pathway may be a better strategy when developing AMD therapies [60].

### 3.5. Bioactive Lipids and AMD

Lipids have a very crucial role in the pathogenesis of AMD i.e., in the form of drusen (lipid-containing deposits), the role of bioactive lipids as preventive therapy in AMD was explored by various groups. Studies have revealed that omega-3 polyunsaturated fatty acids (ω-3-PUFA) such as eicosapentaenoic acid from dietary sources such as fish and nuts reduced the risk of early AMD [61,62,63,64]. It was observed that in hypoxic stimulation of neovascularization, the increasing ω-3-PUFA tissue levels led to vessel regrowth after injury leading to a decreased avascular zone of the retina in the case of AMD [65]. Furthermore, it has been noted that PUFA-derived mediators, such as neuroprotectin D1, resolvin D1, and resolvin E1, were very effective in stopping neovascularization. The protective benefits of ω-3-PUFAs and their bioactive metabolites were mediated, in part, by the inhibition of tumor necrosis factor-α [65].

Oxidized lipids, especially 7-ketocholesterol have been reported in various studies to play a significant role in inflammation and angiogenesis in various age-related and lifestyle diseases due to their accumulation in various tissues and body fluids [66]. Aging is responsible for the accumulation of 7-ketocholesterol in various ocular tissues, especially the retina and drusen [67]. Recently, 7-ketocholesterol has been demonstrated to contribute to the pathogenesis of AMD by inducing ER stress and MAPK pathways via p38-induced inflammation and cell death [68]. Conversion of Cholesterol to 7-ketocholesterol is postulated to be mediated by a free radical mechanism supported by the ferrous ion (potentially from ferritin and mitochondrial cytochrome c) mediated in the presence of light [69].

## 4. Therapeutic Intervention in AMD

### 4.1. Management of AMD

Due to the lack of therapeutic options for wet AMD before the introduction of anti-VEGF antibodies. Wet AMD although present in 10% of the total population having AMD was the cause of more than 90% of cases of legal blindness [70,71,72,73]. Currently, anti-VEGF therapies have become a gold standard for the therapy of wet AMD but there has not been a very effective therapy for dry AMD until now. However, in the case of dry AMD, numerous curative possibilities are being examined by aiming at (1) preventive approaches, (2) pausing AMD succession, and/or (3) restoration of eyesight. The most essential treatment selection for AMD includes photodynamic therapy (PDT), laser photocoagulation, anti-VEGF injections, combination therapies, and many more treatments [74,75].

Preventive approaches in dry AMD include the usage of drugs with antioxidant potential, cessation of smoking, and maintaining a healthy lifestyle to reduce risk factors such as hypertension and obesity. Some physical approaches such as the use of blue light filtering lenses especially in older populations which has undergone cataract surgery are not clinically established but clinical trials to establish a correlation are being conducted [76]. Another example of physical intervention in AMD is laser-based photocoagulation. It has been demonstrated that this technique leads to a reduction in the drusen area, but the studies were largely inadequate [76]. Intraocular implantable miniature telescope was clinically tested to improve the quality of life for patients with late-stage AMD [72]. It was found to significantly improve vision, but there were also some negative effects, including corneal edema, inflammatory deposits, and elevated intraocular pressure.

In the cases of intermediate stages of dry AMD, the formulations containing a combination of antioxidant compounds have been evaluated as a therapeutic option for pausing the progression of AMD and lowering the chance of vision loss. In addition to that some recent clinical trials have reported the use of drugs targeting complement cascade which is a major cause of dry AMD. In other approaches, stem cells have been used to replace retinal cells to restore vision and are currently used as investigative therapy and are currently in clinical testing [77,78,79]. Anti-VEGF treatment remains the backbone of treatment for wet AMD; bispecific antibodies that target both PDGF and VEGF have additionally been identified [45,46,80]. In the following section, various types of drugs that are used in the therapy of AMD or under clinical trials will be discussed in detail especially for drugs targeting angiogenesis, inflammation, and oxidative stress in the management of AMD.

#### 4.1.1. Anti-VEGF Agents

As discussed earlier, angiogenesis plays a very important role in the transition of dry-state AMD to Wet AMD contributing to a loss of vision leading to poor quality of life in a geriatric population. Hence, reducing angiogenesis is the major goal of the treatment. The first FDA-approved therapy for targeting VEGF was bevacizumab. It is an antibody for metastatic colorectal cancer, which is a generalized anti-VEGF antibody used for the off-label in wet AMD. Furthermore, pegaptanib sodium, an aptamer was the first therapy that has been permitted by the USFDA for the treatment of wet AMD and targeting VEGF 165 isoform which is responsible for neovascularization in wet AMD. Ranibizumab (a humanized monoclonal fab fragment) is another anti-VEGF agent targeting all VEGF A isoforms approved of wet AMD in 2006 based upon outcomes of ANCHOR and Marina trials with benefits sustaining up to 24 months [81]. Various anti-VEGF and their biosimilars are enlisted in Table 1.

As of late 2019, cellular angiogenesis is inhibited by a humanized single-chain antibody fragment called brolucizumab, which binds to all VEGF isoforms with great affinity. The HAWK and HARRIER phase III clinical trials (NCT02307682 and NCT02434328) have provided evidence to support the US FDA’s approval of its usage in clinics. These trials were able to show that, after 48 weeks of therapy, brolucizumab (6 mg) outperformed aflibercept (2 mg) in terms of how well it corrected visual function while maintaining a similar safety profile in terms of side effects. The duration of the effect was longer up to 3 months as suggested by current studies [82].

Additional experimental methods to target VEGF that are presently undergoing clinical studies include Abicipar-pegol, a non-monoclonal antibody anti-VEGF that was developed based on ankyrin repeat proteins and has just finished phase III clinical trials [27]. KSI-301, a humanized anti-VEGF monoclonal antibody coupled to a high molecular weight phosphorylcholine-based biopolymer, is another investigational treatment undergoing phase III trials. Conjugated biopolymer is crucial in decreasing immunogenicity and lengthening the duration of residency in the eye, which prolongs the therapeutic effects. This greatly enhanced patient compliance by reducing the discomfort associated with frequent anti-VEGF intravitreal injections [15].

Another family of anti-VEGF macromolecules is recombinant fusion proteins (RFPs), which function as decoy receptors for PGF and VEGF isoforms [83]. RFPs have the two highest affinity domains against VEGFR isoforms and the Fc part of a completely monoclonal antibody [84]. There is one USFDA-approved molecule Aflibercept that demonstrated higher binding affinity than anti-VEGF mAb, another molecule in this class is conbercept in clinical trials for diabetic macular edema [85].

#### 4.1.2. Anti-VEGF Biosimilars

The European Union and the United States are about to lose their patent protection for bevacizumab, ranibizumab, and aflibercept, or their patents have already expired. Because of this, anti-VEGF biosimilar drugs that imitate the effects of well-known treatments but lack the active components of generic treatments have been developed. Bevacizumab biosimilars are currently available as Mvasi and Zirabev, with ONS-5010 in development. FYB201, Xlucane, SB11, PF582, and Razumab are all ranibizumab biosimilars in development [86]. While aflibercept is still under patent, anti-VEGF biosimilars such as MYL1701, ALT-L9, FYB203, and CHS2020 are in the queue [27].

**Table 1 medicina-60-01647-t001:** FDA-approved anti-VEGF therapies and their biosimilars.

Anti-VEGF Drugs	Structure	Target/Mechanism of Action	MW (kDa)	Half-Life in Days	FDA Approval	Side Effects
Pegaptanib Macugen^®^	Pegylated neutralizing RNA aptamer	Inhbits VEGF-A_165 isoforms	50	Human: 10	2004	Anterior chamber inflammation, blurred vision, cataract, conjunctival hemorrhage, hypertension, increased intraocular pressure, punctate keratitis, reduced visual acuity
Ranibizumab Lucentis^®^	Recombinant humanized mAb IgG1 [87]	Inhbits All isoforms of VEGF-A	48	Rabbit: 2.88 Human: 9	2006	Conjunctival Hemorrhage, eye pain, vitreous floaters, and increased IOP
Byooviz™ Ranibizumab biosimilar	Recombinant humanized IgG1 kappa isotype monoclonal antibody fragment	Inhbits All isoforms of VEGF-A	48	Human: 9	2021 Got interchangeability status in 2023	Conjunctival Hemorrhage, eye pain, vitreous floaters, and increased IOP
CIMERLI™ (ranibizumab-eqrn)		Inhbits All isoforms of VEGF-A	48	Human: 9	2022	Conjunctival Hemorrhage, eye pain, and increased IOP
Ximluci	Recombinant humanized IgG1 kappa isotype monoclonal antibody fragment	Inhbits All isoforms of VEGF-A	48	Human: 9	Approved by EMA in 2022	Retinal detachment, resulting in flashes of light with floaters progressing to a temporary loss of sight or a clouding of the lens (cataract)
Bevacizumab Avastin^®^	Recombinant humanized full-length mAb IgG1	Inhbits All isoforms of VEGF-A	149	Rabbit: 4.3 Human: 4.9	Off-label use for AMD	Epistaxis, headache, hypertension, rhinitis, proteinuria, taste alteration, dry skin, rectal hemorrhage, lacrimation disorder, and exfoliative dermatitis.
Aflibercept Eylea^®^	Fusion of the second domain of VEGFRs 1 and the third domain of VEGFR 2 to the Fc portion of human IgG1	Inhbits All isoforms of VEGF-A and B	115	Rabbit: 3.63 Human: 11	2011	EYLEA caused conjunctival hemorrhage, eye pain, cataracts, vitreous detachment, vitreous floaters, and increased intraocular pressure.
Yesafili	Fusion of the second domain of VEGFRs 1 and the third domain of VEGFR 2 to the Fc portion of human IgG	Attaches to VEGF A and B, to placental growth factor as decoy	7 days	Rabbit: 3.63 Human: 11	EMA authorization in 2023	Aflibercept caused conjunctival hemorrhage, eye pain, cataracts, vitreous detachment, vitreous floaters, and intraocular pressure increased.
Brolucizumab Beovu^®^	region of human IgG1 IG Fv Fragment and single-chain antibody fragment (scFv)	Inhbits All isoforms of VEGF-A	26	Cynomolgus monkey: 2.4 days (in ocular compartments)	2019	Blurred vision, cataracts, conjunctival hemorrhage, eye pain, and vitreous floaters

Abbreviations: FDA (Food and Drug Administration US), MW (Molecular weight), EMA (European Medicine Agency), IG (Immunoglobulin), mAB (Monoclonal Antibody), IOP (Intraocular pressure).

#### 4.1.3. Other Antiangiogenic Biotherapeutics

Volociximab is a monoclonal antibody under development that targets 51 integrins, a crucial protein involved in angiogenesis. The therapy is already in advanced stages for cancer and has recently got approval from the USFDA for Phase I trials.

#### 4.1.4. Bispecific Antibodies

Recently, the US FDA authorized Genentech’s bispecific antibody dual targeting biotherapeutic, faricimab-svoa for AMD treatment. By simultaneously and independently binding to both VEGF-A and angiopoietin-2 (Ang-2), the medication inhibits angiogenesis at the molecular level. It is thought that the anti-Ang 2 impact makes the blood vessels less sensitive to VEGF-A’s effects and increases vascular stability [27].

Efdamrofusp alfa (code: IBI302) is a bispecific fusion protein that can neutralize both C3b/C4b and VEGF isoforms. In a laser-induced CNV mice model, Efdamrofusp alfa showed higher efficacy than anti-VEGF monotherapy following intravitreal injection. Macrophage invasion and M2 macrophage polarization were shown to be further inhibited by dual suppression of VEGF and complement activation. NCT03814291, a phase 1 dose-escalation clinical trial, was started based on preclinical results. The first set of data indicated that nAMD patients tolerated efdamrofusp alfa well [88].

#### 4.1.5. Small Molecules

Oral kinase inhibitors are being explored in the therapy because inhibition of tyrosine kinase results in blocking the action of VEGF and PDGF. Recently, a phase II trial (Apex trial) demonstrated a comparable ocular outcome along with anti-VEGF injections, but the study was stopped in between due to limited tolerability due to gastrointestinal and hepatobiliary adverse events [89,90].

Verteporfin (Visudyne^®^ liposomal formulation) PDT is the first formulation to successfully counteract the loss of visual acuity in patients with AMD and sub-foveal choroidal neovascularisation (CNV) associated with AMD [91]. The TAP Study Group (which lasted a year) released the findings from two randomized clinical studies using verteporfin and PDT for individuals with CNV [92]. Either verteporfin or a placebo was instilled into the patients at random, and either way, the verteporfin was activated or the placebo-treated patients received a fake treatment using laser light. After a year of follow-up, a visual acuity benefit was seen for the entire trial group allocated to verteporfin therapy, and it was even more pronounced for classic sub-foveal lesions (where the area of classic CNV was at least 50% of the size of the entire lesion).

Squalamine lactate is an antimicrobial aminosterol compound discovered in the tissues of dogfish sharks that blocks angiogenesis through an enduring intracellular pathway [93]. Caveolae are bulb-shaped invaginations in the plasma membrane that allow the drug to enter into activated endothelial cells. Subsequently, the drug attaches itself to calmodulin and “chaperones”, which inhibits angiogenesis. Squalamine lactate was first tried as an intravenous therapy, but with its quick clearance and difficult administration, this approach was canceled. These shortcomings led to the development of a new ocular formulation which was used for phase II trials but eventually failed to meet primary endpoints.

Retinal scarring and angiogenesis can be stopped by FGF-2 inhibitors. In some preclinical models, an FGF-2 aptamer (RBM-007) prevented (Figure 3) laser-induced CNV, retinal fibrosis, and pharmacokinetic investigations in rabbit vitreous indicated high and sustained profiles in comparison to approved anti-VEGF medications [94,95,96]. The SELEX approach (systematic evolution of ligands by exponential enrichment) was applied to select the FGF-2 aptamer (a short single-stranded nucleic acid molecule [97]. The idea lies in the fact that short oligonucleotides can fold into distinct three-dimensional structures when they encounter a target, binding the target with a high degree of specificity and affinity. Comprising 37 nucleotides with a high affinity for FGF2 (KD 20 pM), RBM-007 has a strong pharmacokinetic profile (T_1/2_ > 24 h) due to its conjugation with 40-kDa polyethylene glycol (PEG) and significantly modified ribose 2′ locations to withstand ribonucleases.

In a phase II TOFU trial (Ribomic USA Inc. NCT04200248), four monthly intravitreal injections of RBM-007, are being administered either alone or in conjunction with aflibercept. Additionally, intravitreal injections of RC-28, a recombinant dual decoy receptor IgG1 Fc-fusion protein that inhibits both VEGF and FGF-2, have been used in Phase II clinical trials (NCT04270669) by RemeGen Ltd. In addition to targeting the formation of retinal scars, these recently developed FGF-2-targeted therapeutic drugs may offer a significant alternative for patients who were not sensitive to anti-VEGF.

#### 4.1.6. Ocular Gene Therapies

Ocular gene therapies are used for targeting a variety of ocular disorders but are considered very important in the case of retinal disorders [75]. Currently, there are five gene therapies (i.e., Luxturna, ADVM-022, RGX-314, GT-005, and HMR59) that are in trials for AMD [98]. All these therapies are based on the utilization of AAV-based vectors for gene delivery [99,100]. The advantages such as its inability to replicate by itself and its nonpathogenic nature make it a system of choice for gene therapy. AVV-based gene therapy can be done via genomic and nongenomic integration. Non-genomic integration is when the plasmid is not able to replicate itself and will not change cellular DNA, so will require multiple injections to sustain the effect, but it can be used for the cells that are mature and have less mitotic activity. AAV vectors can be employed for controlling genetic integration which integrates the gene at a specific locus on chromosome 19. This approach can be used to reverse genetic pathologies permanently with a single injection. Another important advantage is that the AAV virus can evade the host immune system leading to less or mild reactions on injection.

Retinal pigment epithelium-specific 65 KDa protein, or RPE65, is a retinoid isomerohydrolase gene. In 2008, three independent research groups reported a successful subretinal injection of this expression vector using AAV to improve vision in people with Leber’s congenital amaurosis, an inherited form of blindness [101]. This led to the development of Luxturna (voretigene neparvovec-rzyl), the first gene therapy medicine for the eye to receive FDA approval. Perhaps more significantly, though, was the fact that this early success of gene therapy in the eye helped to revitalize the field in general following a grave failure linked to a patient’s death, which prompted a closer examination of the biology of virus vectors. The foundation for the creation of gene treatments for AMD was laid by these experiments.

ADVM-022 was produced by Adverum Biotechnologies as a gene therapy to treat diabetic macular edema and wet AMD. Clinical trials in Phase 1 and Phase 2 are presently being conducted to study it. Wet AMD is caused by CNV formation, which is supported by VEGF overactivity. Using an AAV.7m8 capsid to deliver a codon-optimized cDNA encoding an aflibercept-like protein, ADVM-022 is intended to decrease VEGF activity. Based on preclinical studies showing the impact for up to 30 months, this gene therapy may lessen the requirement for repeated intravitreal injections, hence reducing the treatment burden for patients [98,102].

Wet AMD might furthermore be treated with AAV8-associated gene therapy, such as RGX-314 from REGENXBIO Inc. A monoclonal antibody fragment produced by RGX-314 resembles ranibizumab, a well-known anti-VEGF medication. A humanized monoclonal antibody fragment called ranibizumab binds to human VEGF-A to reduce CNV. By injecting RGX-314 subretinal or suprachoroidal route, it may be possible to produce anti-VEGF antibodies steadily and lessen the need for repeated intravitreal injections [98].

Gyroscope Therapeutics’ FOCUS trial used GT-005 gene therapy to treat patients with dry AMD, which makes up the majority of AMD cases and for whom there are presently no proven treatments [103]. One theory for the cause of dry AMD is the dysregulation of the complement system, which is exacerbated by an overactive alternative route. Utilizing an AAV2 vector, GT-005 introduces a plasmid construct that codes for a typical Complement Factor I (CFI) protein, which functions as a natural complement system inhibitor. This construct could express itself constitutively following a single dose. Phase 1 (FOCUS) and Phase 2 (HORIZON and EXPLORE) clinical trials are presently testing GT-005 for safety and efficacy.

The HMR59 gene therapy from Hemera Biosciences was designed for people with dry AMD and neovascular age-related macular degeneration (nAMD) [104]. It uses an AAV2 vector to express the protein CD59, which prevents the formation of MAC complexes during late-stage complement system activation. HMR59 upregulates CD59 expression on RPE cells in an attempt to guard against the complement cascade thought to be responsible for macular neovascularization. In clinical trials, HMR59 is administered intravitreally for seven days following a single anti-VEGF injection to treat wet AMD. To ascertain its effectiveness and safety, HMR-1002, a Phase 1 clinical trial, is presently testing it. HMR59 is also being studied in the HMR-1001 experiment for dry AMD. The therapy has been reported to be well tolerated, with three of seventeen subjects experiencing mild virtutis that resolved with topical steroids. At the highest dose, there was a 23% reduction in geographic atrophy, and no therapy eyes were changed to wet AMD during the 18-month follow-up.

#### 4.1.7. Antioxidant Therapies

Age-Related Eye Disease Studies (ARDES) was a clinical study to assess the usefulness of antioxidant supplements for the prevention and to reduce disease progression in a clinical setting for AMD. The supplements such as Beta-carotene (15 mg), vitamin E (400 IU), vitamin C (500 mg), cupric acid (2 mg), and zinc oxide (80 mg) were evaluated and results demonstrated an inhibitory effect on the progression of AMD [105,106].

In the ARDES2 trial, it was found that carotene was linked with higher incidences of a group having a smoking history. This observation led to the replacement of carotenoids with lutein + zeaxanthin supplementation in the original ARDES formulation [107]. OT-551 is an anti-inflammatory and antioxidant molecule that protects rats’ RPE cells from light-induced degeneration [108]. An open-label phase II trial (*n* = 11; NCT00306488) employed 0.45 percent OT-551 topical solution to participants with bilateral GA in a randomly assigned eye. There were no significant contrary events reported among the 11 participants. After two years, the benefits of treatment were considered minimal or non-existent [107].

#### 4.1.8. Role of Anti-Inflammatory Drugs on Inhibition of Angiogenesis in Experimental AMD

Anti-inflammatory drugs have been used widely in cases of ocular disorders as well as in cases of ocular inflammation. These drugs include corticosteroids for instance Triamcinolone acetonide, NSAIDs, and immunosuppressive drugs such as methotrexate (Figure 4).

Corticosteroids have been reported to demonstrate antiangiogenic, anti-inflammatory, and antifibrinogenic activity. These drugs were amongst the first therapy to be tried against CNV. Apart from their anti-inflammatory properties, corticosteroids have been shown to decrease choroidal endothelial cell permeability and inhibit VEGF expression [109].

A study reported the use of combination therapy with Dexamethasone and either photodynamic therapy or anti-VEGF therapy to treat CNV lesions in AMD. It has been shown that this triple therapy stabilizes visual acuity in patients with wet AMD and reduces the number of anti-VEGF injections needed [110].

Triamcinolone acetonide (TA) is a long-acting corticosteroid reported to be clinically used in the therapy of wet AMD either as periocular or intravitreal therapy. In a study, it was found that intravitreal administration of TA resulted in enhanced visual sharpness after 1 and 3 months of the therapy, and vision gain was maximum in patients having RPE attachment [111]. The effects produced by TA are temporary and reversal has been reported in trials. In the case of combination therapy, TA with PDT was reported to stabilize the vision for 2 years in addition to reduced vascular leakage. Corticosteroids are associated with side effects, most notably increased IOP and cataract formation. When compared to dexamethasone, TA has comparable efficacy; however, due to its prolonged duration in the eye, TA is more likely to cause adverse effects such as elevated IOP and cataracts [112].

**Figure 4 medicina-60-01647-f004:**
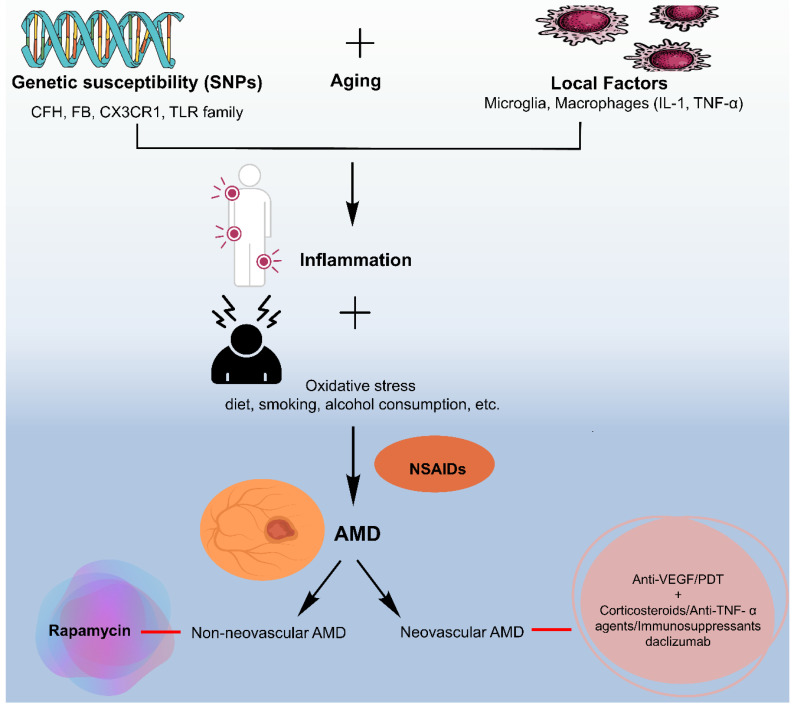
Role of anti-inflammatory therapy on AMD. Modified from [112].

Nonsteroidal anti-inflammatory drugs (NSAIDs) are commonly employed to reduce inflammation and avoid miosis following cataract surgery, thereby easing postoperative pain. It was observed that NSAIDs can reduce the risk of cystoid macular edema and lessen the pain and inflammation brought on by allergic conjunctivitis and keratitis [113]. NSAIDs are strong inhibitors of the COX pathway, one of the arachidonic acid metabolic processes that results in decreased prostaglandin synthesis. COX-2 plays an important role in CNV due to its presence in vascular endothelium and RPE. Macrophages and proinflammatory cytokines significantly increase their expression. It has been demonstrated that COX-2 controls the expression of VEGF and its receptors, which are involved in the formation of CNV in AMD. Corticosteroids, which inhibit both LPO and COX, have greater and more potent anti-inflammatory effects than NSAIDs; however, the use of NSAIDs does not increase intraocular pressure. It is commonly known that topical NSAIDs can cause hypersensitivity and ocular toxicity.

Immunosuppressant for example methotrexate (MTX, a dihydrofolate reductase inhibitor) was tested in clinical settings for the therapy of AMD. In the patients who did not respond to anti-VEGF medication, MTX alone was competent to alleviate symptoms. In another study, a combination therapy of methotrexate and anti-VEGF therapy (bevacizumab) showed an improvement in vision sharpness with no additional side effects [114].

## 5. Strategic Drug Delivery Approaches

### 5.1. Routes of Ocular Drug Delivery

Drug administration to the eye comprises topical, intravitreal, subconjunctival, and retrobulbar routes [115,116,117]. Other less popular routes are peribulbar, sub-tendon, and intracameral routes [118]. Apart from the topical route, different routes require highly sophisticated techniques and can only be performed in clinical settings [119,120].

Intravitreal injection of formulations enables the access of drugs directly at vitreous humor and pars plana to the posterior part of the eye, bypassing the barriers encountered if delivery is done through a topical route. Diseases affecting the retina, like AMD, and diabetic retinopathy, are treated using anti-VEGF solutions like Lucentis^®^ (Ranibizumab) and Avestin^®^ (Bevacizumab) using this route, and steroids like solutions (Kenalog-40), depots (suspensions and implants, e.g., TRIESENCE™ and OZURDEX^®^) for management of posterior uveitis, exudative macular degeneration, macular edema (diabetic and vasculo-occlusive) and pseudophakic cystoid macular edema leading to improvement of symptoms and prevention of vision impairment [121]. The process of elimination is usually initiated along the retina or the anterior part of aqueous humor after the equilibrium. The most common risks involved with this route are ocular hemorrhage (10%), endophthalmitis (0.019 to 1.6%), intraocular inflammation (1.4–2.9%), and hematogenous retinal detachment (0–0.67%) [15,16].

The subconjunctival route (SC) can increase the absorption and contact time of the drugs. In contrast with the intravitreal injection, the sub-conjunctival routes are less invasive and permit the dose to proceed along the sclera and gain its entry into the posterior region of the eye.

The drug is injected via orbital fascia and eyelid in the retrobulbar route, thus allowing the direct entry of the dose in the retrobulbar space. This process may affect the optic nerve and is suitable for delivering the anesthetic agents. In the peribulbar route, the drug is injected above and below the orbit; however, this technique is less effective than another route. This route is used for delivering anesthesia in case of cataract surgery. When the drug is injected into the cavity present between the scleral parts and the tenon’s capsule, it is termed the sub-tendon route. This route is safe, quick, and effective as compared to the peribulbar and retrobulbar routes and delivers local anesthetics [122,123,124]. The Intracameral (IC) route provides the drug to the anterior parts of the eye via injection, preferred to deliver the antibiotics. The IC route is chosen to arrest endophthalmitis, which generally occurs after post-operative and cataract surgery [125].

Ocular pharmacokinetics is a critical aspect of designing drug delivery systems for ocular infection and helps to describe absorption, distribution, metabolism, and elimination [126,127]. It is further useful for figuring out the drug’s site of action in ocular tissues as well as the duration and concentrations at which it is present in different ocular tissues. The delivery of a drug depends on the route of administration and formulation characteristics. In the next section, ocular pharmacokinetics is discussed according to the different target segments of the eye.

### 5.2. Intravitreal Pharmacokinetics

Delivery of the drug at the posterior tissue segment plays a challenge due to various barriers like blood-ocular barriers. The success rate for drug delivery at the posterior segment is directly proportional to the drug that reaches the action site. The amount of drug at the posterior part depends on the formulation and route of administration [128]. The factors affecting pharmacokinetics after intravitreal injection are distribution and clearance of vitreous humor [129]. As vitreous humor has a high viscosity, a pore size of 500 nm, and is negatively charged. The distribution of drugs and nanoparticles may happen primarily by diffusion, and convection (movement of aqueous humor synthesized by the ciliary activities to the retina via the vitreous humor) performs a very minimal role in the transport of drugs [130,131]. The vitreal distribution may additionally be dependent upon the electrostatic charges of the molecule/nanoparticle [132].

The movement of small molecules [133] is faster as compared to larger molecular-weight drugs [134]. Clearance from vitreous humor occurs via posterior and anterior clearance routes depending on lipophilicity and molecular weight of the drug, and t_1/2_ in vitreous humor follows the order dexamethasone < ceftizoxime < octreotide acetate [125]. Nomoto et al. in 2009 investigated the pharmacokinetics of the drug bevacizumab in the eye of rabbits by using three different modes of administration, i.e., intravitreal injection, subconjunctival injection, and eye contact drops. The delivery of 1.25 mg/0.05 mL of bevacizumab was done into the eye by using eye drops, subconjunctival injection, and finally, by using intravitreal injection and the highest concentration of bevacizumab is found in the iris/ciliary body and retina/choroid (109,192.6 and 93,990 ng/g) via intravitreal injection [135]. The diffusion process of drugs beyond the area of vitreous humor, which further gains its entry into the posterior chamber of the eye, is one of the two pathways for the elimination process of drugs from the anterior segment, posterior segment, and vitreous part. This process is influenced by the clearance rate, blood flow in the uvea, and aqueous humor turnover rate. In the area of the posterior route drugs gain access in the retina via permeation process but are cleared rapidly by the flow of blood in choroidal parts.

## 6. Formulations for Therapy of AMD with Preclinical/Clinical Studies

As discussed above, the major route for AMD therapy is the intravitreal route. Topical and periocular routes are used less frequently. The main aim of developing formulation is for spatial positioning of a drug in the vicinity of the tissue where the drug must exhibit the activity [136]. Another focus is to prolong the therapy by employing formulation development so that the need for frequent intravitreal injection is avoided [137]. These objectives can be achieved by making depots of the drug(s) in delivery systems to control the spacio-temporal delivery of the drug.

Likewise, the therapy for AMD relies on macromolecular therapeutic agents such as antibodies and aptamers. As well, the therapy target is usually the retina and macular region. Hence, the intravitreal route is the first practical choice for the administration of drug(s) for the treatment of AMD. Moreover, the intravitreal route is associated with drawbacks for example an increase in intraocular pressure, intraocular inflammation, and risk of bacterial endophthalmitis after intravitreal injection of bevacizumab, ranibizumab, or aflibercept. In addition to that protein aggregation in the case of protein therapeutics is a lingering issue and contaminations from prefilled syringes, especially of silicone oil which is commonly used as lubricant in prefilled syringes utilized for intravitreal injections.

Hence, ocular drug delivery systems should be fabricated to deliver drug(s) to target areas or tissues of an eye by overcoming ocular barriers, improving treatment efficiency and drug stability, prolonging drug retention time and reducing the dosing frequency, allowing various drug amalgamations, and decreasing drug-related adversative measures and improve patient compliance [138,139,140]. Conventional delivery techniques, which have been employed extensively in clinical settings and have produced some therapeutic outcomes, include topical eye drops, conjunctival and scleral administration, intracameral injection, intravitreal injection, retrobulbar injection, and systemic administration [141]. However as was previously indicated, ocular barriers present a major obstacle to therapies in terms of getting to the intended site and remaining there long enough. Thus, the bioavailability of ocular therapeutics is often reduced, usually less than 5% [142]. Likewise, the ocular barriers limit the practical utilization of conventional drug delivery systems for the posterior chamber of the eye [143]. Hence, cautious, and target-specific techniques are needed to overcome ocular barriers for effective therapeutic moiety delivery and a prolonged therapeutic effect. So, developments of sustained release and nanotechnology-based ocular drug delivery systems have become more important for effective treatment administration to the posterior ocular area. The pros and cons of various delivery systems are shown in Figure 5.

Over the last few decades, nanotechnology has had a significant impact and presents a great opportunity to develop new ocular delivery methods for the safe and efficient administration of anti-VEGF medicines to treat AMD. The emergence of nanotechnology has led to rapid advancements in the field of ocular medication delivery and offers novel therapeutic approaches for treating various eye disorders [142,144]. The ability to cross ocular barriers, increase transcorneal permeability, extend drug residence time, decrease drug degradation, lower dosage frequency, enhance patient compliance, achieve sustained/controlled release, drug targeting, and gene delivery are just a few of the many benefits that nanocarriers have over traditional drug administration [145,146].

Numerous drug delivery systems can be applied for ocular purposes including nanoemulsions, nanomicelles, nanosuspensions, nanoparticles, nanofibers, microemulsions, dendrimers, liposomes, nanowafers, niosomes, contact lenses, microneedles, nanocrystals, hydrogels, and innovative gene therapy approaches have been developed and covered elsewhere in other publications [147,148,149,150,151]. Liposomes, Solid lipid NP (SLN), and nanostructured lipid carriers (NLC) are lipid-based nanoformulations that can be utilized to encase active molecules [152].

For a literature review on “drug delivery”, “nanomedicine”, “nanoparticles”, and “nanotechnology” based strategies for AMD, the reader can refer to the list of the review papers for detailed literature on advances in drug delivery systems for AMD treatment as mentioned in Table 2. Some important nano delivery systems are discussed below.

Figure 6 shows the various delivery systems that can be employed to deliver the active components to treat eye disorders. Moreover, unlike traditional drug delivery techniques, the physicochemical properties of nanoformulations, such as their size, shape, and surface charge, may influence how efficiently they penetrate ocular tissues [153]. Several drug delivery systems that can be used to develop to treat AMD are listed in Table 3.

### 6.1. Polymeric Nanoparticles

Polymeric nanoparticles with sizes ranging from 10–1000 nm are utilized to deliver drugs to the retina. Non-toxic, non-immunogenic, biodegradable, bio-adhesive, and biocompatible polymers such as Poly (D, L-lactide-co-glycolide) (PLGA), chitosan, PEG, albumin, and hyaluronic acid (HA) can be used to prepare polymeric nanoparticles [154]. Drug delivery systems that are bio- or mucoadhesive (like chitosan) have a longer half-life in the mucosal layer, which increases their bioavailability.

The scientist conducted a comparative study of the reduction in the size of lab-induced CNV by bevacizumab, poly(ethylene glycol) (PEG)-bevacizumab conjugate (b-PEG), and poly(lactic-co-glycolic acid) (PLGA)-encapsulated bevacizumab (b-PLGA) in rat eyes [155]. Using laser photocoagulation, Bruch’s membrane in the eyes ruptured. Except for those in group 3, all treatment subgroups showed a decrease in CNV area and results indicated that bevacizumab formulations were successfully designed while maintaining their potent antiangiogenic properties.

**Table 2 medicina-60-01647-t002:** List of review papers on “drug delivery”, “nanomedicine”, “nanoparticles”, and “nanotechnology” based strategies for the treatment of AMD.

Title	Highlights of Article	Year of Publication	References
Long-acting intraocular Delivery strategies for biological therapy of age–related macular degeneration”	Physiological and anatomical barriers to drug delivery. Prospects for biological therapeutics. Development of drug delivery methodologies.	2019	[156]
Recent theranostic paradigms for the management of Age-related macular degeneration	Application of printing 3D and AI to manage AMD. Possibilities of research in therapy.	2020	[157]
Nanotechnology for Age-Related Macular Degeneration	Development of nano-drug delivery systems and gene therapy strategies. Novel targeting strategies and the potential application of delivery methods.	2021	[158]
Nanotechnology: revolutionizing the delivery of drugs to treat age-related macular degeneration	AMD biology and the pathophysiology. Successes and limitations of available therapies. Novel therapeutics.	2021	[159]
Therapeutic Approaches for Age-Related Macular Degeneration	Recent challenges encountered in the treatment of different forms of AMD. Innovative nanoformulations, 3D bioprinting, and techniques to monitor the progress.	2022	[160]
Novel Approaches in the Drug Development and Delivery Systems for Age-Related Macular Degeneration	Approaches for the treatment of AMD. Novel drug delivery systems and route of administration for AMD.	2023	[161]
Advanced nanomedicines for the treatment of age-related macular degeneration	Latest pre-clinical treatment options in ocular drug delivery to the retina. Explores the advantages of nanoparticle-based therapeutic approaches for AMD.	2024	[162]
Age-Related Macular Degeneration—Therapies and Their Delivery	Different types of nanocarriers developed for the topical ocular delivery system. Important treatment options for AMD.	2024	[163]

Further, researchers synthesized a thermos-reversible gel that contained sunitinib malate poly(lactic-co-glycolic acid) nanoparticles [164]. When gel-loaded and blank nanoparticles were tested against drug solution at 10 μM and 20 μM concentrations, respectively, the viability of ARPE-19 cells was found to be greater than 90%. The synthesized formulations demonstrated a better reduction of VEGF activity against drug solution, higher absorption, and increased anti-angiogenic action, according to the wound scratch assay, cellular uptake, and VEGF expression levels. Ultimately, they concluded that a thermo-reversible gel based on nanoparticles loaded with sunitinib could be utilized to treat neovascular AMD.

### 6.2. Lipid-Based Nanoformulations

#### Liposome

Liposomes are lipid vesicles that have a diameter of 0.1–10 microns, good biocompatibility, good encapsulating ability, and the ability to regulate drug release [158]. Due to the possibility of surface modification and control drug release because of the number and composition of lipid bilayers, liposomes are generally utilized in ocular disease research. For instance, redox-sensitive N-acetylcysteine-loaded smart liposomes were developed and enhancement in the expression of antioxidant genes in retinal pigment epithelial (hESC-RPE) cells was observed [165]. The focused treatment of retinal degeneration now has a new avenue to pursue thanks to this innovative delivery method. According to Joseph et al., ranibizumab-loaded DPPC–DPPG liposomes constantly released the medication compared to other formulations [166].

A dual-modified ophthalmic liposome (Penetratin hyaluronic acid-liposome/Conbercept, PenHA-Lip/Conb, a topical delivery) was constructed to non-invasively penetrate the ocular barrier and deliver anti-VEGF therapeutic agents to the targeted intraocular tissue [167]. Penetratin (Pen) serves as the ocular penetration enhancer, while hyaluronic acid (HA) acts as the retina-targeting ligand.

2-deoxy-D-glucose was loaded into liposomes modified with the RGD peptide (RGD@LP-2-DG) to design a biocompatible nanomedicine delivery system [168]. Excellent in vitro and in vivo inhibitory effects on neovascularization were established by RGD@LP-2-DG, which demonstrated good targeting performance towards endothelial cells.

The FDA has approved Visudyne^®^, a liposomal formulation used in PDT for nAMD that contains photosensitizer verteporfin [169]. Egg yolk phosphatidylglycerol (egg yolk PG) and dimyristoyl phosphatidylcholine (DMPC), together with additional additions including lactose, palmitic acid, ascorbic acid, and dibutyl hydroxytoluene, are the components of hydrophobic verteporfin, which is intended for intravenous injection and is encapsulated in liposomes. Following intravenous (IV) treatment, the photosensitizer is stored in the unusual and leaky blood vessels within the eye. Infrared light exposure causes photosensitizer to create cytotoxic ROS (free radicals), which inhibit CNV in the eye by selectively obstructing newly formed blood vessels. This reduces bleeding and exudation.

### 6.3. Metallic Nanoparticles

IVT injections have been utilized to introduce and access deep layers of the retina and choroidal region using metal nanoparticles (gold, silver, and iron oxide) as nanotherapeutics and for non-diagnostic purposes [149]. In certain conditions that are exclusively linked to retinal degeneration, photoreceptor precursor (PRP) transplantation is thought to be a useful method of improving visual acuity [170]. Analyzing cell survival in the host retina is a crucial component of this kind of treatment. Chemla et al. developed AuNPs to enhance the tracking of transplanted PRPs [171]. After being labeled with photothermic AuNPs, the cells of long-Evans pigmented rats (4–8 weeks old) were transplanted into the vitreous and sub-retinal region using a gauge needle. The results confirmed AuNPs’ ability to improve human retinal cell therapy.

### 6.4. Non-Metallic Nanoparticles

A new class of carbonaceous nanomaterials known as carbon quantum dots (or CDs) has drawn a lot of interest because of its superior water dispersion, low cytotoxicity, ease of synthesis, incredibly small size, excellent fluorescence, and capacity to enter cells and tissues [172]. These characteristics have led to a great deal of research on CDs as potential bioimaging probes, effective medicine transporters, and disease diagnostic tools. One possible treatment option for eye disorders is functionalized CDs.

### 6.5. Clinical Investigations of Formulations

A clinical investigation was carried out in 2015 on 67 individuals with early AMD. The study involved the continuous intake of carotenoid or their formulations (Lutein, zeaxanthin, meso-zeaxanthin, Macuhealth, Macushield) [173]. Macular pigment (MP) increased significantly (*p* < 0.05) at three years. They concluded that MP could be improved with a range of supplements in the early stages of AMD. However, meso-zeaxanthin can provide additional benefits in terms of contrast sensitivity and panprofile enhancement.

A high-dose lutein/zeaxanthin supplement was studied to see how it affected the levels of skin carotenoid (SC) and macular pigment optical density in healthy individuals [174]. This is an open-label, prospective, single-arm trial. The 16 Japanese subjects ranged in age from 26 to 57. For 16 weeks, the subjects received a supplement that included additional antioxidants (vitamins C, E, zinc, and copper) together with 20 mg of lutein and 4 mg of zeaxanthin per day. The findings showed that a high dose of lutein/zeaxanthin supplementation was beneficial for MPOD volume and SC levels without causing significant side effects.

**Table 3 medicina-60-01647-t003:** List of nanoformulations developed in preliminary stages or preclinical stages as ocular drug delivery to treat AMD or related disorders.

Drug(s)/Active Components	Formulation(s)	Route of Administration	Outcome(s)	References
Sunitinib	Liposomes	Intravitreal injection	Inhibitory effect on mice in a laser-induced CNV	[175]
Bevacizumab	Liposomes	Intravitreal injection	More effective as anti-angiogenesis in nAMD, good permeation via the cell model barrier, long-term stability improved	[176]
microRNA-150 and quercetin	Lipid NPs	--	Have a potent effect on the fundus, preventing CNV for up to two weeks in a rat model without endangering the retina.	[177]
Angiopoietin 1	PLGA NPs	intravenous injection	Successfully lessen the leaking of neovascularization	[178]
Mesenchymal stem cell (MSC)	exosomes	In vitro study in ARPE-19 cells	Using the Nrf2/Kepa1 signaling pathway regulation, exosomes shield RPE cells from oxidative damage.	[179]
Siglecs (sialic-acid-binding immunoglobulin-type lectins)	PolySialic acid-nanoparticles	Intravitreal injection	PolySia-NPs reduced the size of neovascular lesions	[180]
Cerium nitrate	Cerium nanoparticles	Topical eye drops	By reducing VEGF and raising PEDF levels, therapy reduced laser-induced choroidal neovascular lesions in mice.	[181]
Dexamethasone and bevacizumab	PLGA and polyethylenimine NPs	Intravitreal injection	Showed good anti-angiogenic effect on HUVEC cells, Enhanced inhibitory effect on VEGF secretion	[182]
Bevacizumab	Chitosan NPs in hyaluronic acid	Ocular implant	Sustained release	[183]
Aflibercept and Dexamethasone	Micro- and nanoparticle hydrogel	--	Sustained release up to 224 days	[184]
Everolimus	Nanomicelles with Soluplus^®^	Topical	Enhance permeation of drug via cornea	[185]
Ovalbumin	PLGA NP loaded bilayer microneedle	Ex vivo study on porcine sclera	Sustained release of protein and bypasses scleral barrier	[186]
Dasatinib	Spray-dried PLGA particles	Intravitreal injection	Prolonged release and notable suppression of collagen matrix contraction.	[187]
Fenofibrate	PLGA NPs	Intravitreal injection	Improved retinal vascular leakage, inhibited retinal leukostasis, controlled VEGF overexpression, and decreased injection frequency	[188]
Bevacizumab	Albumin PLGA NPs.	Vitreous injection in New Zealand albino rabbits	Sustained-release formulation of bevacizumab and extended for about 8 weeks	[189]
Bevacizumab	Carbon nanovesicles	Intravitreal injection	increased bioavailability, sustained release	[41]
Sirolimus	Chitosan functionalized PLGA NPs	In vitro, and ex vivo studies	NPs penetrated more to scleral tissue, less cytotoxicity	[190]
Cerium oxide and Melanin	ceria-coated melanin-PEG nanoparticles (CMNPs)	Intravitreal injection	new monotherapy intended to protect the RPE and photoreceptors in AMD	[191]

The development of eye drops could serve as a monotherapy, enable longer intervals between IVT injections of standard therapy, or be used as a follow-up to IVT injections to stabilize active disease. By lowering the risk of infection from IVT injections and boosting patient and caregiver adherence and compliance through fewer clinic visits, drops would be convenient. A once-daily eye drop suspension for nAMD with PAN-90806 (PanOptica), a tiny chemical that binds VEGFR2 and inhibits tyrosine kinase activity, is being developed (NCT03479372) [192]. Some trials with altered formulations confirmed punctate keratopathy due to off-target suppression of corneal EGFR [193]. Some of the clinical trials or preclinical experiments of various delivery systems for the treatment of AMD, nAMD, or retinal degeneration are summarized in Table 4.

**Table 4 medicina-60-01647-t004:** Some clinical trials of delivery systems for AMD treatment.

Drug(s)/Actives	MOA	Delivery System	Use(s)	Suggested Route of Administration	Stage(s)	References
Bevacizumab	VEGF inhibitor	Intracapsular drug ring	Exudative AMD	During cataract surgery	Preclinical	[194]
Ixoberogene Soroparvovec	anti-VEGF	Gene therapy	AMD	Intravitreal	phase 1 study	[195]
Voretigene Neparvovec		Gene therapy	Retinal degeneration	Subretinal injection/oral	Successful phase I to III studies	[196]
Pan-9080622	VEGF/FGF/tyrosine kinase inhibitor	Eyedrops	Exudative AMD	Topical	Phase I/II	[197]
Ranibizumab	VEGF inhibitor	Liposome	Exudative AMD	Subconjunctival	Preclinical	[166]
Ranibizumab	VEGF inhibitor	Refillable port delivery system	Exudative AMD	Trans-scleral implantation	Phase III trials	[198]
GB-102 (Sunitinib Malate)	tyrosine kinase inhibitor	Bioerodable polymeric nanoparticles	neovascular AMD	Intravitreal depot injection	Phase I/II trials	[199]
LHA-510 (Acrizanib)	Tyrosine kinase-VEGF receptor inhibitor	eyedrops	exudative AMD	Topically	Failed Phase II trial	[200]
Bevacizumab	VEGF inhibitor	Liposome	Exudative AMD	Intravitreal injection	Preclinical	[201]
Dexamethasone, + Aflibercept	Anti-inflammatory and VEGF inhibitor	Polymeric nanoparticles	Wet AMD		Release time of 224 days	[184]

Eye drops that can administer TAK-593, a VEGF receptor tyrosine kinase inhibitor, to the posterior portion of the eye were designed by Mori et al. in 2023 [202]. Based on the characteristics and the emulsion’s viscosity, xanthan gum was chosen as a viscosity booster. The concentration of the formulation and the inclusion of viscosity enhancers improved the transport of TAK-593 to the posterior eye. TAK-593 emulsion eye drops demonstrated the same angiogenesis-suppression efficacy as anti-VEGF antibody intravitreal injection in the laser-induced CNV model.

## 7. Conclusions and Prospects for the Future

The prominent reason for vision loss in aging people in developed or developing nations is AMD. Environmental, genetic, and other risk factors are associated with this complex chronic inflammatory disorder. With repeated doses of anti-VEGF agents (Lucentis, Avastin, Eylea, and Macugen), AMD can be treated in many patients. However, these repeated injections can lead to intraocular hemorrhage, inflammation of the eyes, endophthalmitis, the elevation of intraocular pressure, vitreous/retinal detachment, and other problems.

So, AMD patients need less harmful and more efficient pharmacologic treatment to restore their eyesight. The drug bioavailability that is efficacy of active compounds can be increased by either augmenting corneal permeability with penetration enhancers, or extending retention time by mucoadhesive polymer, and nanoformulations/nanocarriers. Additionally, the current research must aim to decrease the frequency of intravenous injections of treatments and enhancement of long-term results by investigating new non-invasive treatment approaches and long-acting intraocular extended-release delivery systems. To facilitate topical ocular distribution, various nanocarriers for instance liposomes, polymeric nanoparticles, and lipid nanoformulations are being developed by researchers. When applied topically, these topical ocular nanocarriers can increase the solubility of hydrophobic agent(s) and may enhance bioavailability. Consequently, the use of nanocarriers as topical ocular delivery systems can also offer a viable therapeutic approach and safe means of treating posterior segment ocular illnesses, such as AMD. Hence, it can be supposed that a wide range of therapeutic and diagnostic technologies, including exosome stem cell therapy, tissue engineering, gene therapy, and topical delivery based on nanotechnology, are anticipated to have a major impact on AMD diagnosis and therapy in the future.

## Figures and Tables

**Figure 1 medicina-60-01647-f001:**
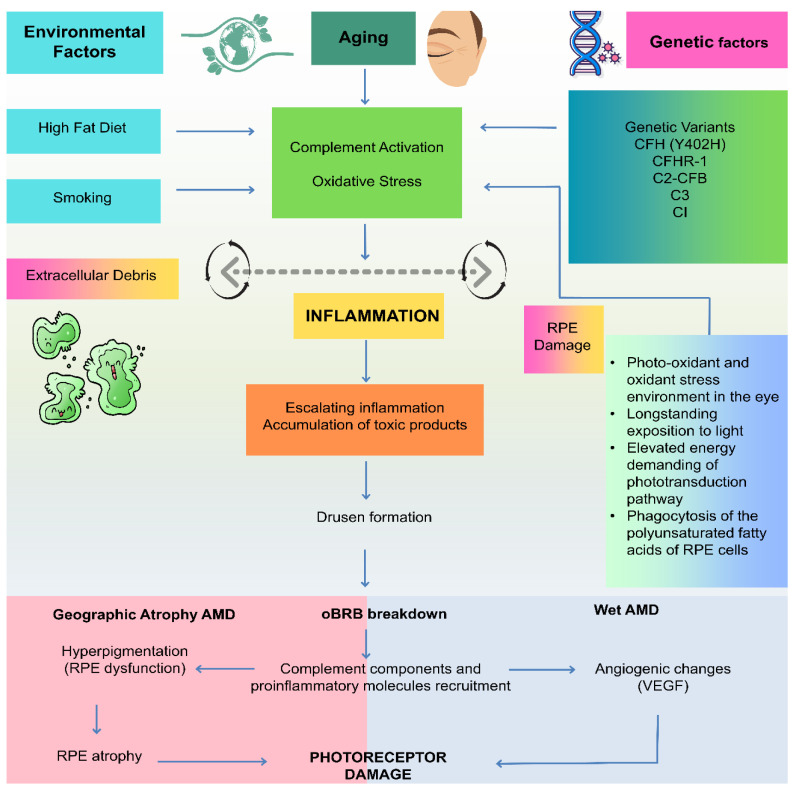
Interplay of aging, genetics, and environmental factors on the pathophysiology of AMD. retinal pigment epithelium (RPE), blood-retinal barrier (BRB), Complement factor H (CFH) gene, and CFH-related gene 1 (CFHR-1).

**Figure 2 medicina-60-01647-f002:**
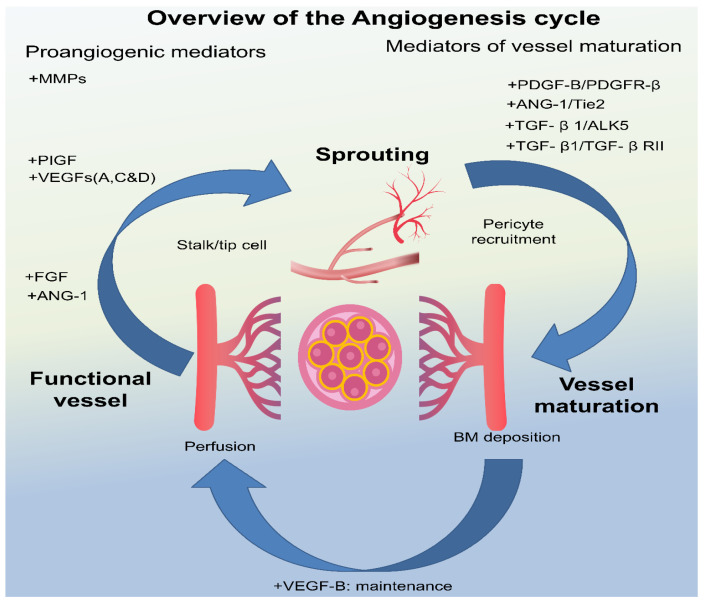
Role of molecular mediators in Angiogenesis.

**Figure 3 medicina-60-01647-f003:**
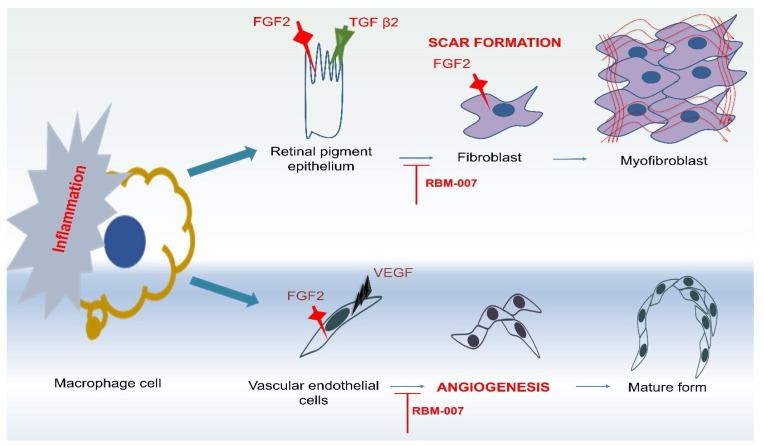
Diagram illustrating FGF2’s dual function in the development of fibrotic scars and angiogenesis in the retinal macrophage cell. RBM-007 can inhibit FGF2 and VEGF to prevent scar formation and angiogenesis.

**Figure 5 medicina-60-01647-f005:**
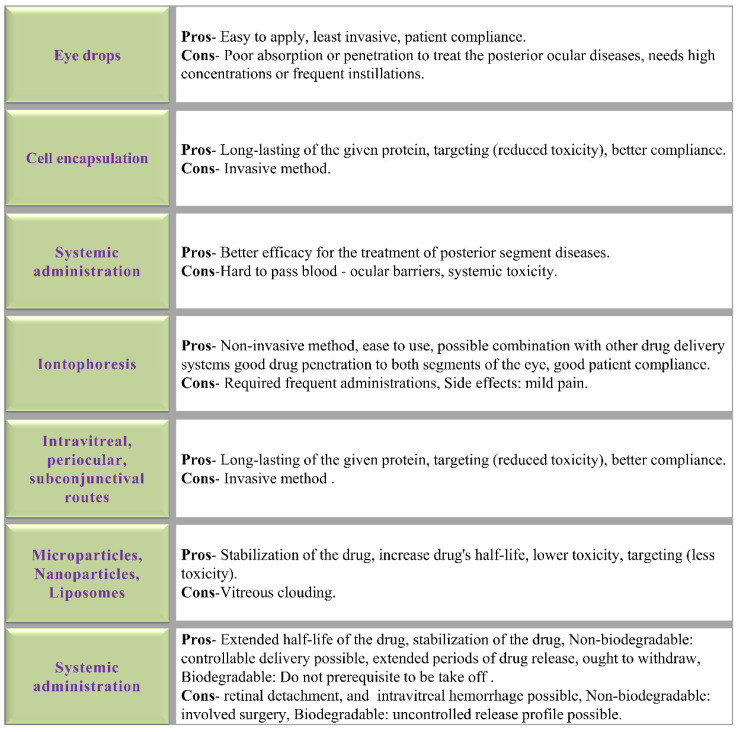
Various plausible ocular drug delivery with their pros and cons.

**Figure 6 medicina-60-01647-f006:**
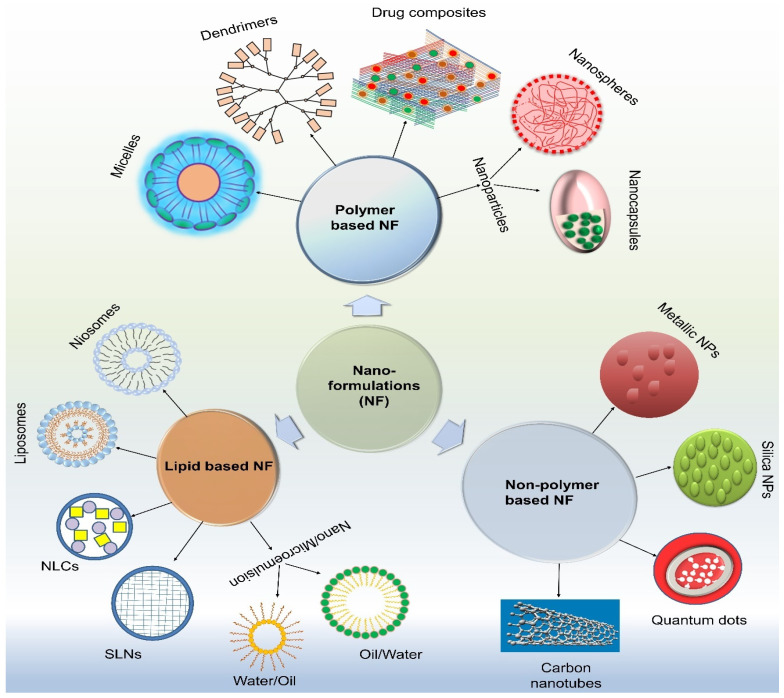
Diagrammatic representation of various probable nanoformulations that can be applied as an ocular drug delivery system for treatment or diagnostic purposes.

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
