# Peer review of "Age-Related Macular Degeneration (AMD): Pathophysiology, Drug Targeting Approaches, and Recent Developments in Nanotherapeutics"

_medicina, 2024, doi:10.3390/medicina60101647_

Round 1

Reviewer 1 Report

Comments and Suggestions for Authors

Please see the comments in attached documnet 37

Comments on the Quality of English Language

Minor editing of English language is required.

Author Response

Response to Reviewers’ Comments

Ms. No: medicina-3217239

The manuscript entitled: Age-related Macular Degeneration (AMD): Pathophysiology, Drug Targeting Approaches, and Recent Developments in Nanotherapeutics

Journal Name: Medicina, MDPI.

Authors thankful to reviewers and editors for moral encouraging comments and remarks.

Reviewer #1 Comments

The paper reviewed the pathophysiology, Drug Targeting Approaches, and Recent Developments in Nanotherapeutics of Age-related Macular Degeneration (AMD). The review is interesting, however, requires considerable editorial revision. Requires grammar and English proficiency check.   

The paper reviewed the pathophysiology, Drug Targeting Approaches, and Recent Developments in Nanotherapeutics of Age-related Macular Degeneration (AMD). The review is interesting, however, requires some editorial revision. Requires grammar and English proficiency check.

Comment 1: Abstract

The acronym anti-VEGF should be written in full considering it was first time use.

Response : Thank you for your useful suggestion. The authors have been made the correction in the revised manuscript. Line 20, Page 1.

Give examples of the developments in Nanotherapeutics

Response : Thank you for your valuable suggestion. The authors have been made the correction in the revised manuscript. Line 25, Page 1.

Comments 2: Introduction

Don’t start a paragraph with acronym (Line 41)

Response : Thank you for your valuable suggestion. The authors have made the corrections in the revised manuscript. Line 37, Page 2.

Add more to paragraph one. Describe the process and theory of Aging.

Response : Thank you for your helpful suggestion. We agree to add the information. The authors have made the corrections in the revised manuscript. Lines 34-52, Pages 1-2.

Justify this claim in Line 37-38: “AMD in elderly patients has arisen as one of the obvious triggers of blindness remarkably in developed, and prevalent in developing countries” with facts and examples. Cite source of information.

Response : Thank you for your useful suggestion. We agree to add the information. The authors have made the corrections in the revised manuscript. Lines 90-104, Pages 3.

Give few examples to this factors listed in Line 41-42. Eg: “AMD can be described as a chronic, complex disease that is influenced by multiple genetic variants like…….; environmental factors like……”

Response : Thank you for your useful suggestion. We agree to add the information. The authors have made the corrections in the revised manuscript. Lines 71-74, Pages 2.

Comment 3: Other observations

Dedicate a section or sub-heading describing concept, etiology, types and symptoms of Age-related Macular Degeneration.

Response : Thank you for your valuable comment. We agree to add the information. The authors have made the corrections in the revised manuscript. Lines 70-85, Pages 2-3.

Under section 2: Prevalence and Risk factors of AMD. Please kindly do a holistic review on spread of prevalence giving examples both for developing and developed countries. Although global perspective is captured.

Response : Thank you for your valuable comment. We agree to add the information. The authors have made the corrections in the revised manuscript. Lines 90-104, Page 3.

Please cite sources to all figures generated. 

Response : Thank you for your valuable comment. We agree to add the information. The authors have made the corrections in the revised manuscript.

Table 1 have some missing data. Please check. Add two columns describing the mechanism of action and possible side effect for each Anti-VEGF drugs. Add key at the bottom of the Table describing each acronym used. Eg FDA, MW, EMA, IG etc.

Response : Thank you for your valuable comments. We agree to add the information. The authors have made the corrections in the revised manuscript. Lines 394-396, Pages 10-11.

Line 469. Write the acronym ARDES in full

Response : Thank you for your valuable suggestions. We agree to add the information. The authors have made the corrections in the revised manuscript. Lines 515, Page 14.

Information on the advances and developments in Nanotherapeutics is scanty. Please improve.

Response : Thank you for your valuable suggestions. We agree to add the information. The authors have made the corrections in the revised manuscript. Lines 683-686, Lines 692-789, Pages 20-23.

Recast the conclusion. Too lengthy.

Response : Thank you for your valuable suggestions. We agree to make the correction. The authors have made the corrections in the revised manuscript. Lines 822-844, Page 25-26.

Comments on the Quality of English Language-Requires grammar and English proficiency check.

Response : Thank you for your worthy comment to improve the manuscript. We agree to correct it. We have done the English proofread as per your suggestions. The authors have made the corrections in the revised manuscript.

Reviewer 2 Report

Comments and Suggestions for Authors

The paper reviewed the pathophysiology, Drug Targeting Approaches, and Recent Developments in Nanotherapeutics of Age-related Macular Degeneration (AMD). The review is interesting, however, requires considerable editorial revision. Requires grammar and English proficiency check.   

The paper reviewed the pathophysiology, Drug Targeting Approaches, and Recent Developments in Nanotherapeutics of Age-related Macular Degeneration (AMD). The review is interesting, however, requires some editorial revision. Requires grammar and English proficiency check.

Abstract

·         The acronym anti-VEGF should be written in full considering it was first time use.

·         Give examples of the developments in Nanotherapeutics

Introduction

·         Don’t start a paragraph with acronym (Line 41)

·         Add more to paragraph one. Describe the process and theory of Aging

·         Justify this claim in Line 37-38: “AMD in elderly patients has arisen as one of the obvious triggers of blindness remarkably in developed, and prevalent in developing countries” with facts and examples. Cite source of information.

·         Give few examples to this factors listed in Line 41-42. Eg: “AMD can be described as a chronic, complex disease that is influenced by multiple genetic variants like…….; environmental factors like……”

Other observations

·         Dedicate a section or sub-heading describing concept, etiology, types and symptoms of Age-related Macular Degeneration

·         Under section 2: Prevalence and Risk factors of AMD. Please kindly do a holistic review on spread of prevalence giving examples both for developing and developed countries. Although global perspective is captured.

·         Please cite sources to all figures generated. 

·         Table 1 have some missing data. Please check. Add two columns describing the mechanism of action and possible side effect for each Anti-VEGF drugs.

·         Add key at the bottom of the Table describing each acronym used. Eg FDA, MW, EMA, IG etc

·         Line 469. Write the acronym ARDES in full

·         Information on the advances and developments in Nanotherapeutics is scanty. Please improve

·         Recast the conclusion. Too lengthy.

Comments on the Quality of English Language

Requires grammar and English proficiency check.

Author Response

Response to Reviewers’ Comments

 Ms. No: medicina-3217239

 The manuscript entitled: Age-related Macular Degeneration (AMD): Pathophysiology, Drug Targeting Approaches, and Recent Developments in Nanotherapeutics

Journal Name: Medicina, MDPI.

Authors thankful to reviewers and editors for moral encouraging comments and remarks.

Reviewer #2 Comments

Well, this was in interesting review.  Very detailed, and probably different sections were written by different authors.  This is most obvious in the stylistic differences in the writing (i.e.: one authors likes to use the word "also" a Iot, another describes random studies etc...) suggest that a singular author reads the entire manuscript and cleans up the stylistic issues.

Comments 1: Please use passive voice in the entire manuscript - so not need to cite the actual authors and describe the study (e.g.: lines 74-75)

Response : Thank you for your suggestions and we agree with comment. We have made corrections to the revised manuscript.  

Comments 2: There are missing definitions and biological errors/mis-statements in this manuscript.  

line 37: define AMD (leave no acronym unexplained)

Response : Line 57, Page 2, Age-related macular degeneration (AMD). The correction has been made in the revised manuscript.

line 94: RPE, Bruch's membrane, and the choroid (not the neurosensory retina)

Response : Thank you for pointing out this. We have corrected the revised manuscript. Line 129-130, Page 4.

line 193 - what is a pericycle?  Pericyte?

Response : Thank you for pointing out the error. The correction has been made in the revised manuscript. Line 229, page 6. It is the pericyte.

line 300 - implants of telescopes?  Rather a large apparatus to fit into an eye (telescopes are used to watch stars in t sky).

Response : Thank you for pointing out the error and we agree to your pertinent query. The authors regret such a mistake. The appropriate correction has been made in the revised manuscript. Line 336-337, Page 9. We used the below given reference (72).

  1. Savastano A, Ferrara S, Sasso P, et al. Smaller-Incision new-generation implantable miniature telescope: Three-months follow-up study. European Journal of Ophthalmology. 2023:11206721231212545.

line 394: caveolae are not tiny holes, they are bulb-shaped invaginations in the plasma membrane.

Response : Thank you for pointing out the error and we agree to correct it. The correction has been made in the revised manuscript. Lines 432-433, page 12.

line 580 - posterior chamber of the eye (correct), line 611: posterior eye (how many eyes are there?  Spiders have anterior and posterior eyes....)

Response : Thank you for the pertinent comment. The corrections have been made in the revised manuscript. Line 630, page 18, and Line 662, page 19. It is the posterior chamber of the eye.

line 594: you mean "treatment" not "cure"

Response : Thank you for a pertinent suggestion. The correction has been made in the revised manuscript. It will be treatment. Line 665, page 19.

line 606 - intracameral?

Response : Thank you for the pertinent query. An intracameral injection is usually of an antibiotic into the anterior chamber of the eyeball to prevent endophthalmitis

Comments 3:  There are sections where sentences don't quite make sense:

line 63-65: regions of aging populations and then more in Asia (as the population ages).  What are the original regions?  Tortuous sentence.

Response : Thank you for the pertinent comment. The corrections have been made in the revised manuscript. Line 91-95, page 3.

  1. Xu, X., Wu, J., Yu, X., Tang, Y., Tang, X. and Shentu, X., 2020. Regional differences in the global burden of age-related macular degeneration. BMC Public Health20, pp.1-9.

line 66-68: white population vs dark-skinned.  Is this a commentary on race (Caucasian being white and African being darker) or does this take into account the pale skins of northern Asians vs. eh darker skins of equatorial Asians?

Response : Thank you for your valuable query. Frank et al. reviewed the literature and have presumed that AMD is much more common in white persons than in persons of black African inheritance. Line 97-99, page 3.

  1. Frank, R.N., Puklin, J.E., Stock, C. and Canter, L.A., 2000. Race, iris color, and age-related macular degeneration. Transactions of the American Ophthalmological Society, 98, p.109.

line 157-161 - jumbled sentence - need clarification and chopping into smaller sentences. One idea per sentence, please.

Response : Thank you for your suggestion. The authors agree to make the corrections in the revised manuscript. Lines 187-192, Page 5.

line 288 - the word is "until" not "till" which is a cash register.

Response : Thank you for your suggestion. The authors have corrected the revised manuscript. Line 325, Page 8.

lines 299-302 - yep, that telescope section need clarification

Response 3: Thank you for your pertinent query. It will be the ‘implantable miniature telescope’. The correction has been made in the revised manuscript. Lines 336-337, Page 9.

  1. Savastano A, Ferrara S, Sasso P, et al. Smaller-Incision new-generation implantable miniature telescope: Three-months follow-up study. European Journal of Ophthalmology. 2023:11206721231212545.

lines 400-403 - run on sentence

Response : Thank you for pertinent comment. The authors agree to correct it. The correction has been made in the revised manuscript. Lines 431-435, page 12.

lines 528-535 - run on sentences.  Please fix.

Response : Thank you for your valuable suggestion. The authors agree to correct this. The correction has been made in the revised manuscript. Lines 567-571, page 16-17.

lines 556-557: why is that there?  Delete next sentence.

Response : Thank you for your valuable suggestion. The authors agree to change it. The authors have been made the changes in the revised manuscript.

lines 590-591: "depots for by formulation????"

Response 3: Thank you for your pertinent comment. The authors agree to change it. It will be ‘depots of the drug(s)’. The authors have made the corrections in the revised manuscript. Lines 641, page 18.

lines 600-604 -run on sentences

Response : Thank you for your valuable suggestion. The authors agree to change it. The corrections have been made in the revised manuscript. Line 642-645, page 18.

line 731: not important, rather " currently most successful"

Response : Thank you for your valuable suggestion. The authors agree to change it. The corrections have been made in the revised manuscript.

line 733 - "vetoed"?  Not quite the right word

Response : Thank you for your pertinent comment. The authors agree to change it. The corrections have been made in the revised manuscript. Line 828, page 25

line 736: "unharmed"  Nope, still not quite the right word

Response 3: Thank you for your valuable comment. The authors agree to change it. The corrections have been made in the revised manuscript. Line 831, page 25.

Comments 4: Missing references

Response : Thank you for giving the suggestions. The authors agree to correct these suggestions point to point. The corrections have been made in the revised manuscript.

Tables 1 and 3 are not cited in the text.  There is no table 2.  There is no figure 4

Response : Thank you for highlighting the errors. The authors agree to correct these errors. The corrections have been made in the revised manuscript. Table 1: Line 362-363, page 9, Table 2: 687, page 20, Figure 4: Line 534, Pages 15-16.

line 68

Response : The following references are added in the revised manuscript for line 68.

  1. Wong WL, Su X, Li X, et al. Global prevalence of age-related macular degeneration and disease burden projection for 2020 and 2040: a systematic review and meta-analysis. The Lancet Global Health. 2014;2(2):e106-e116.

  1. Xu X, Wu J, Yu X, et al. Regional differences in the global burden of age-related macular degeneration. BMC Public Health. 2020;20:1-9.

lines 352-359,

Response : The following references are added in the revised manuscript for lines 352-359.

  1. Formica, M.L., Awde Alfonso, H.G. and Palma, S.D., 2021. Biological drug therapy for ocular angiogenesis: Anti‐VEGF agents and novel strategies based on nanotechnology. Pharmacology Research & Perspectives9(2), p.e00723.
  2. Alfonso, H.G.A., Paz, M.C., Palma, S.D. and Formica, M.L., 2023. Advances in nanotechnology-based anti-VEGF agents for the management of ocular angiogenesis. In Nanotechnology in Ophthalmology(pp. 247-262). Academic Press.
  3. Khachigian, L.M., Liew, G., Teo, K.Y., Wong, T.Y. and Mitchell, P., 2023. Emerging therapeutic strategies for unmet need in neovascular age-related macular degeneration. Journal of Translational Medicine21(1), p.133.

lines 361-363

Response : The following references are added in the revised manuscript for lines 361-363.

  1. Kaiser, S.M., Arepalli, S. and Ehlers, J.P., 2021. Current and future anti-VEGF agents for neovascular age-related macular degeneration. Journal of Experimental Pharmacology, pp.905-912.
  2. Sarkar, A., Sodha, S.J., Junnuthula, V., Kolimi, P. and Dyawanapelly, S., 2022. Novel and investigational therapies for wet and dry age-related macular degeneration. Drug discovery today27(8), pp.2322-2332.

line 515,

Response : The following reference is added in the revised manuscript for line 515.

  1. Rigas, B., Huang, W. and Honkanen, R., 2020. NSAID-induced corneal melt: Clinical importance, pathogenesis, and risk mitigation. Survey of ophthalmology65(1), pp.1-11.

line 542, line 545, line 546, line 548,

Response : The following references are added in the revised manuscript for line 542, line 545, line 546, line 548.

  1. Ilochonwu, B.C., Van Der Lugt, S.A., Annala, A., Di Marco, G., Sampon, T., Siepmann, J., Siepmann, F., Hennink, W.E. and Vermonden, T., 2023. Thermo-responsive Diels-Alder stabilized hydrogels for ocular drug delivery of a corticosteroid and an anti-VEGF fab fragment. Journal of Controlled Release361, pp.334-349.
  2. Ghasemi Falavarjani, K. and Nguyen, Q.D., 2013. Adverse events and complications associated with intravitreal injection of anti-VEGF agents: a review of literature. Eye27(7), pp.787-794.
  3. Gomez-Lumbreras, A., Ghule, P., Panchal, R., Giannouchos, T., Lockhart, C.M. and Brixner, D., 2023. Real-world evidence in the use of Bevacizumab in age-related macular degeneration (ArMD): a scoping review. International Ophthalmology43(12), pp.4527-4539.

line 599

Response : The following references are added in the revised manuscript for line 599.

  1. Spruill, M.L., Maletic-Savatic, M., Martin, H., Li, F. and Liu, X., 2022. Spatial analysis of drug absorption, distribution, metabolism, and toxicology using mass spectrometry imaging. Biochemical pharmacology201, p.115080.
  2. Mandal, A., Pal, D., Agrahari, V., Trinh, H.M., Joseph, M. and Mitra, A.K., 2018. Ocular delivery of proteins and peptides: Challenges and novel formulation approaches. Advanced drug delivery reviews126, pp.67-95.

 Comments 5: missing pieces

lines 53-54: there needs to be connecting statement between the pervious paragraph and the subsequent one - is this the first major review?  How does this review add to the field?

Response : Thank you for your valuable suggestion. The authors agree to make the changes.

lines 82 and 88: refer to "initial stages" and advanced stages". Previous sections were about "dry" vs. "wet" Please clarify.

Response : Thank you for your valuable comments. The dry AMD is also called the initial stage of AMD and when it is converted to wet AMD, then it is called as an advanced stage of AMD. So, we have mentioned the dry or initial stage, wet or advanced stage of AMD in the manuscript.

in section 4.1.6 - Why is Luxturna not discussed?  The others are discussed.

Response : Thank you for your pertinent comment. The authors have added the information in the revised manuscript. Lines 472-480, Pages 13-14.

line 495: is Triamcinolone acetonide a corticosteroid?  Not mentioned in the introductory paragraph. Please use acronym in the remaining parts of that paragraph.

Response : Thank you for your pertinent comment and suggestions. The authors have made the corrections in the revised manuscript. Line 531, 543, 545, 546, page 15.

Comments 5: Comments on the Quality of English Language

The multiple authors have a rather combative approach to the English language.  It is a difficult language to learn as there are many odd nuances.  Several weird sentence structures, word choices and missing punctuation marks abound.  Please find a singular proof reader to check all that, removing all the excessive  sequential "also's" and "therefores".  

Response : Thank you for your worthy comment to improve the manuscript. We agree to correct it. We have done the English proofread as per your suggestions. The authors have made the corrections in the revised manuscript.

Reviewer 3 Report

Comments and Suggestions for Authors

Well, this was in interesting review.  Very detailed, and probably different sections were written by different authors.  This is most obvious in the stylistic differences in the writing (i.e.: one authors likes to use the word "also" a Iot, another describes random studies etc...) suggest that a singular author reads the entire manuscript and cleans up the stylistic issues.

Please use passive voice in the entire manuscript - so not need to cite the actual authors and describe the study (e.g.: lines 74-75)

1) There are missing definitions and biological errors/mis-statements in this manuscript.  

line 37: define AMD (leave no acronym unexplained)

line 94: RPE, Bruch's membrane and the choroid (not the neurosensory retina)

line 193 - what is a pericycle?  Pericyte?

line 300 - implants of telescopes?  Rather a large apparatus to fit into an eye (telescopes are used to watch stars in t sky)

line 394: caveolae are not tiny holes, they are bulb-shaped invaginations in the plasma membrane

line 580 - posterior chamber of eye (correct), line 611: posterior eye (how many eyes are there?  Spiders have anterior and posterior eyes....)

line 594: you mean "treatment" not "cure"

line 606 - intracameral?

2)  There are sections where sentences don't quite make sense:

line 63-65: regions of aging populations and then more in Asia (as the population ages).  What are the original regions?  Tortuous sentence.

line 66-68: white population vs dark-skinned.  Is this a commentary on race (Caucasian being white and African being darker) or does this take into account the pale skins of northern Asians vs. eh darker skins of equatorial Asians?

line 157-161 - jumbled sentence - need clarification and chopping into smaller sentences.  One idea per sentence, please.

line 288 - the word is "until" not "till" which is a cash register.

lines 299-302 - yep, that telescope section need clarification

lines 400-403 - run on sentence

lines 528-535 - run on sentences.  Please fix.

lines 556-557: why is that there?  Delete next sentence.

lines 590-591: "depots for by formulation????"

lines 600-604 -run on sentences

line 731: not important, rather " currently most successful"

line 733 - "vetoed"?  Not quite the right word

line 736: "unharmed"  Nope, still not quite the right word

3) Missing references

Tables 1 and 3 are not cited in the text.  There is no table 2.  There is no figure 4

line 68, lines 352-359, lines 361-363, line 515, line 542, line 545, line 546, line 548, line 599

4) missing pieces

lines 53-54: there needs to be connecting statement between the pervious paragraph and the subsequent one - is this the first major review?  How does this review add to the field?

lines 82 and 88: refer to "initial stages" and advanced stages".  Previous sections were about "dry" vs. "wet"  Please clarify.

in section 4.1.6 - Why is Luxturna not discussed?  The others are discussed.

line 495: is Triamcinolone acetonide a corticosteroid?  Not mentioned in the introductory paragraph.  Please use acronym in the remaining parts of that paragraph.

Comments on the Quality of English Language

The multiple authors have a rather combative approach to the English language.  It is a difficult language to learn as there are many odd nuances.  Several weird sentence structures, word choices and missing punctuation marks abound.  Please find a singular proof reader to check all that, removing all the excessive  sequential "also's" and "therefores".  

Author Response

Response to Reviewers’ Comments

Ms. No: medicina-3217239

The manuscript entitled: Age-related Macular Degeneration (AMD): Pathophysiology, Drug Targeting Approaches, and Recent Developments in Nanotherapeutics

Journal Name: Medicina, MDPI.

Authors thankful to reviewers and editors for moral encouraging comments and remarks.

Reviewer #1 Comments

The paper reviewed the pathophysiology, Drug Targeting Approaches, and Recent Developments in Nanotherapeutics of Age-related Macular Degeneration (AMD). The review is interesting, however, requires considerable editorial revision. Requires grammar and English proficiency check.   

The paper reviewed the pathophysiology, Drug Targeting Approaches, and Recent Developments in Nanotherapeutics of Age-related Macular Degeneration (AMD). The review is interesting, however, requires some editorial revision. Requires grammar and English proficiency check.

Comment 1: Abstract

The acronym anti-VEGF should be written in full considering it was first time use.

Response : Thank you for your useful suggestion. The authors have been made the correction in the revised manuscript. Line 20, Page 1.

Give examples of the developments in Nanotherapeutics

Response : Thank you for your valuable suggestion. The authors have been made the correction in the revised manuscript. Line 25, Page 1.

Comments 2: Introduction

Don’t start a paragraph with acronym (Line 41)

Response : Thank you for your valuable suggestion. The authors have made the corrections in the revised manuscript. Line 37, Page 2.

Add more to paragraph one. Describe the process and theory of Aging.

Response : Thank you for your helpful suggestion. We agree to add the information. The authors have made the corrections in the revised manuscript. Lines 34-52, Pages 1-2.

Justify this claim in Line 37-38: “AMD in elderly patients has arisen as one of the obvious triggers of blindness remarkably in developed, and prevalent in developing countries” with facts and examples. Cite source of information.

Response : Thank you for your useful suggestion. We agree to add the information. The authors have made the corrections in the revised manuscript. Lines 90-104, Pages 3.

Give few examples to this factors listed in Line 41-42. Eg: “AMD can be described as a chronic, complex disease that is influenced by multiple genetic variants like…….; environmental factors like……”

Response : Thank you for your useful suggestion. We agree to add the information. The authors have made the corrections in the revised manuscript. Lines 71-74, Pages 2.

Comment 3: Other observations

Dedicate a section or sub-heading describing concept, etiology, types and symptoms of Age-related Macular Degeneration.

Response : Thank you for your valuable comment. We agree to add the information. The authors have made the corrections in the revised manuscript. Lines 70-85, Pages 2-3.

Under section 2: Prevalence and Risk factors of AMD. Please kindly do a holistic review on spread of prevalence giving examples both for developing and developed countries. Although global perspective is captured.

Response : Thank you for your valuable comment. We agree to add the information. The authors have made the corrections in the revised manuscript. Lines 90-104, Page 3.

Please cite sources to all figures generated. 

Response : Thank you for your valuable comment. We agree to add the information. The authors have made the corrections in the revised manuscript.

Table 1 have some missing data. Please check. Add two columns describing the mechanism of action and possible side effect for each Anti-VEGF drugs. Add key at the bottom of the Table describing each acronym used. Eg FDA, MW, EMA, IG etc.

Response : Thank you for your valuable comments. We agree to add the information. The authors have made the corrections in the revised manuscript. Lines 394-396, Pages 10-11.

Line 469. Write the acronym ARDES in full

Response : Thank you for your valuable suggestions. We agree to add the information. The authors have made the corrections in the revised manuscript. Lines 515, Page 14.

Information on the advances and developments in Nanotherapeutics is scanty. Please improve.

Response : Thank you for your valuable suggestions. We agree to add the information. The authors have made the corrections in the revised manuscript. Lines 683-686, Lines 692-789, Pages 20-23.

Recast the conclusion. Too lengthy.

Response : Thank you for your valuable suggestions. We agree to make the correction. The authors have made the corrections in the revised manuscript. Lines 822-844, Page 25-26.

Comments on the Quality of English Language-Requires grammar and English proficiency check.

Response : Thank you for your worthy comment to improve the manuscript. We agree to correct it. We have done the English proofread as per your suggestions. The authors have made the corrections in the revised manuscript.

Reviewer #2 Comments

Well, this was in interesting review.  Very detailed, and probably different sections were written by different authors.  This is most obvious in the stylistic differences in the writing (i.e.: one authors likes to use the word "also" a Iot, another describes random studies etc...) suggest that a singular author reads the entire manuscript and cleans up the stylistic issues.

Comments 1: Please use passive voice in the entire manuscript - so not need to cite the actual authors and describe the study (e.g.: lines 74-75)

Response : Thank you for your suggestions and we agree with comment. We have made corrections to the revised manuscript.  

Comments 2: There are missing definitions and biological errors/mis-statements in this manuscript.  

line 37: define AMD (leave no acronym unexplained)

Response : Line 57, Page 2, Age-related macular degeneration (AMD). The correction has been made in the revised manuscript.

line 94: RPE, Bruch's membrane, and the choroid (not the neurosensory retina)

Response : Thank you for pointing out this. We have corrected the revised manuscript. Line 129-130, Page 4.

line 193 - what is a pericycle?  Pericyte?

Response : Thank you for pointing out the error. The correction has been made in the revised manuscript. Line 229, page 6. It is the pericyte.

line 300 - implants of telescopes?  Rather a large apparatus to fit into an eye (telescopes are used to watch stars in t sky).

Response : Thank you for pointing out the error and we agree to your pertinent query. The authors regret such a mistake. The appropriate correction has been made in the revised manuscript. Line 336-337, Page 9. We used the below given reference (72).

  1. Savastano A, Ferrara S, Sasso P, et al. Smaller-Incision new-generation implantable miniature telescope: Three-months follow-up study. European Journal of Ophthalmology. 2023:11206721231212545.

line 394: caveolae are not tiny holes, they are bulb-shaped invaginations in the plasma membrane.

Response : Thank you for pointing out the error and we agree to correct it. The correction has been made in the revised manuscript. Lines 432-433, page 12.

line 580 - posterior chamber of the eye (correct), line 611: posterior eye (how many eyes are there?  Spiders have anterior and posterior eyes....)

Response : Thank you for the pertinent comment. The corrections have been made in the revised manuscript. Line 630, page 18, and Line 662, page 19. It is the posterior chamber of the eye.

line 594: you mean "treatment" not "cure"

Response : Thank you for a pertinent suggestion. The correction has been made in the revised manuscript. It will be treatment. Line 665, page 19.

line 606 - intracameral?

Response : Thank you for the pertinent query. An intracameral injection is usually of an antibiotic into the anterior chamber of the eyeball to prevent endophthalmitis

Comments 3:  There are sections where sentences don't quite make sense:

line 63-65: regions of aging populations and then more in Asia (as the population ages).  What are the original regions?  Tortuous sentence.

Response : Thank you for the pertinent comment. The corrections have been made in the revised manuscript. Line 91-95, page 3.

  1. Xu, X., Wu, J., Yu, X., Tang, Y., Tang, X. and Shentu, X., 2020. Regional differences in the global burden of age-related macular degeneration. BMC Public Health20, pp.1-9.

line 66-68: white population vs dark-skinned.  Is this a commentary on race (Caucasian being white and African being darker) or does this take into account the pale skins of northern Asians vs. eh darker skins of equatorial Asians?

Response : Thank you for your valuable query. Frank et al. reviewed the literature and have presumed that AMD is much more common in white persons than in persons of black African inheritance. Line 97-99, page 3.

  1. Frank, R.N., Puklin, J.E., Stock, C. and Canter, L.A., 2000. Race, iris color, and age-related macular degeneration. Transactions of the American Ophthalmological Society, 98, p.109.

line 157-161 - jumbled sentence - need clarification and chopping into smaller sentences. One idea per sentence, please.

Response : Thank you for your suggestion. The authors agree to make the corrections in the revised manuscript. Lines 187-192, Page 5.

line 288 - the word is "until" not "till" which is a cash register.

Response : Thank you for your suggestion. The authors have corrected the revised manuscript. Line 325, Page 8.

lines 299-302 - yep, that telescope section need clarification

Response 3: Thank you for your pertinent query. It will be the ‘implantable miniature telescope’. The correction has been made in the revised manuscript. Lines 336-337, Page 9.

  1. Savastano A, Ferrara S, Sasso P, et al. Smaller-Incision new-generation implantable miniature telescope: Three-months follow-up study. European Journal of Ophthalmology. 2023:11206721231212545.

lines 400-403 - run on sentence

Response : Thank you for pertinent comment. The authors agree to correct it. The correction has been made in the revised manuscript. Lines 431-435, page 12.

lines 528-535 - run on sentences.  Please fix.

Response : Thank you for your valuable suggestion. The authors agree to correct this. The correction has been made in the revised manuscript. Lines 567-571, page 16-17.

lines 556-557: why is that there?  Delete next sentence.

Response : Thank you for your valuable suggestion. The authors agree to change it. The authors have been made the changes in the revised manuscript.

lines 590-591: "depots for by formulation????"

Response 3: Thank you for your pertinent comment. The authors agree to change it. It will be ‘depots of the drug(s)’. The authors have made the corrections in the revised manuscript. Lines 641, page 18.

lines 600-604 -run on sentences

Response : Thank you for your valuable suggestion. The authors agree to change it. The corrections have been made in the revised manuscript. Line 642-645, page 18.

line 731: not important, rather " currently most successful"

Response : Thank you for your valuable suggestion. The authors agree to change it. The corrections have been made in the revised manuscript.

line 733 - "vetoed"?  Not quite the right word

Response : Thank you for your pertinent comment. The authors agree to change it. The corrections have been made in the revised manuscript. Line 828, page 25

line 736: "unharmed"  Nope, still not quite the right word

Response 3: Thank you for your valuable comment. The authors agree to change it. The corrections have been made in the revised manuscript. Line 831, page 25.

Comments 4: Missing references

Response : Thank you for giving the suggestions. The authors agree to correct these suggestions point to point. The corrections have been made in the revised manuscript.

Tables 1 and 3 are not cited in the text.  There is no table 2.  There is no figure 4

Response : Thank you for highlighting the errors. The authors agree to correct these errors. The corrections have been made in the revised manuscript. Table 1: Line 362-363, page 9, Table 2: 687, page 20, Figure 4: Line 534, Pages 15-16.

line 68

Response : The following references are added in the revised manuscript for line 68.

  1. Wong WL, Su X, Li X, et al. Global prevalence of age-related macular degeneration and disease burden projection for 2020 and 2040: a systematic review and meta-analysis. The Lancet Global Health. 2014;2(2):e106-e116.

  1. Xu X, Wu J, Yu X, et al. Regional differences in the global burden of age-related macular degeneration. BMC Public Health. 2020;20:1-9.

lines 352-359,

Response : The following references are added in the revised manuscript for lines 352-359.

  1. Formica, M.L., Awde Alfonso, H.G. and Palma, S.D., 2021. Biological drug therapy for ocular angiogenesis: Anti‐VEGF agents and novel strategies based on nanotechnology. Pharmacology Research & Perspectives9(2), p.e00723.
  2. Alfonso, H.G.A., Paz, M.C., Palma, S.D. and Formica, M.L., 2023. Advances in nanotechnology-based anti-VEGF agents for the management of ocular angiogenesis. In Nanotechnology in Ophthalmology(pp. 247-262). Academic Press.
  3. Khachigian, L.M., Liew, G., Teo, K.Y., Wong, T.Y. and Mitchell, P., 2023. Emerging therapeutic strategies for unmet need in neovascular age-related macular degeneration. Journal of Translational Medicine21(1), p.133.

lines 361-363

Response : The following references are added in the revised manuscript for lines 361-363.

  1. Kaiser, S.M., Arepalli, S. and Ehlers, J.P., 2021. Current and future anti-VEGF agents for neovascular age-related macular degeneration. Journal of Experimental Pharmacology, pp.905-912.
  2. Sarkar, A., Sodha, S.J., Junnuthula, V., Kolimi, P. and Dyawanapelly, S., 2022. Novel and investigational therapies for wet and dry age-related macular degeneration. Drug discovery today27(8), pp.2322-2332.

line 515,

Response : The following reference is added in the revised manuscript for line 515.

  1. Rigas, B., Huang, W. and Honkanen, R., 2020. NSAID-induced corneal melt: Clinical importance, pathogenesis, and risk mitigation. Survey of ophthalmology65(1), pp.1-11.

line 542, line 545, line 546, line 548,

Response : The following references are added in the revised manuscript for line 542, line 545, line 546, line 548.

  1. Ilochonwu, B.C., Van Der Lugt, S.A., Annala, A., Di Marco, G., Sampon, T., Siepmann, J., Siepmann, F., Hennink, W.E. and Vermonden, T., 2023. Thermo-responsive Diels-Alder stabilized hydrogels for ocular drug delivery of a corticosteroid and an anti-VEGF fab fragment. Journal of Controlled Release361, pp.334-349.
  2. Ghasemi Falavarjani, K. and Nguyen, Q.D., 2013. Adverse events and complications associated with intravitreal injection of anti-VEGF agents: a review of literature. Eye27(7), pp.787-794.
  3. Gomez-Lumbreras, A., Ghule, P., Panchal, R., Giannouchos, T., Lockhart, C.M. and Brixner, D., 2023. Real-world evidence in the use of Bevacizumab in age-related macular degeneration (ArMD): a scoping review. International Ophthalmology43(12), pp.4527-4539.

line 599

Response : The following references are added in the revised manuscript for line 599.

  1. Spruill, M.L., Maletic-Savatic, M., Martin, H., Li, F. and Liu, X., 2022. Spatial analysis of drug absorption, distribution, metabolism, and toxicology using mass spectrometry imaging. Biochemical pharmacology201, p.115080.
  2. Mandal, A., Pal, D., Agrahari, V., Trinh, H.M., Joseph, M. and Mitra, A.K., 2018. Ocular delivery of proteins and peptides: Challenges and novel formulation approaches. Advanced drug delivery reviews126, pp.67-95.

 Comments 5: missing pieces

lines 53-54: there needs to be connecting statement between the pervious paragraph and the subsequent one - is this the first major review?  How does this review add to the field?

Response : Thank you for your valuable suggestion. The authors agree to make the changes.

lines 82 and 88: refer to "initial stages" and advanced stages". Previous sections were about "dry" vs. "wet" Please clarify.

Response : Thank you for your valuable comments. The dry AMD is also called the initial stage of AMD and when it is converted to wet AMD, then it is called as an advanced stage of AMD. So, we have mentioned the dry or initial stage, wet or advanced stage of AMD in the manuscript.

in section 4.1.6 - Why is Luxturna not discussed?  The others are discussed.

Response : Thank you for your pertinent comment. The authors have added the information in the revised manuscript. Lines 472-480, Pages 13-14.

line 495: is Triamcinolone acetonide a corticosteroid?  Not mentioned in the introductory paragraph. Please use acronym in the remaining parts of that paragraph.

Response : Thank you for your pertinent comment and suggestions. The authors have made the corrections in the revised manuscript. Line 531, 543, 545, 546, page 15.

Comments 5: Comments on the Quality of English Language

The multiple authors have a rather combative approach to the English language.  It is a difficult language to learn as there are many odd nuances.  Several weird sentence structures, word choices and missing punctuation marks abound.  Please find a singular proof reader to check all that, removing all the excessive  sequential "also's" and "therefores".  

Response : Thank you for your worthy comment to improve the manuscript. We agree to correct it. We have done the English proofread as per your suggestions. The authors have made the corrections in the revised manuscript.